# Dynamic regulation of chromatin accessibility by pluripotency transcription factors across the cell cycle

**Elias T Friman, Cédric Deluz, Antonio CA Meireles-Filho, Subashika Govindan, Vincent Gardeux, Bart Deplancke, David M Suter\***

Institute of Bioengineering, School of Life Sciences, Ecole Polytechnique Fédérale de Lausanne (EPFL), Lausanne, Switzerland

**Abstract** The pioneer activity of transcription factors allows for opening of inaccessible regulatory elements and has been extensively studied in the context of cellular differentiation and reprogramming. In contrast, the function of pioneer activity in self-renewing cell divisions and across the cell cycle is poorly understood. Here we assessed the interplay between OCT4 and SOX2 in controlling chromatin accessibility of mouse embryonic stem cells. We found that OCT4 and SOX2 operate in a largely independent manner even at co-occupied sites, and that their cooperative binding is mostly mediated indirectly through regulation of chromatin accessibility. Controlled protein degradation strategies revealed that the uninterrupted presence of OCT4 is required for post-mitotic re-establishment and interphase maintenance of chromatin accessibility, and that highly OCT4-bound enhancers are particularly vulnerable to transient loss of OCT4 expression. Our study sheds light on the constant pioneer activity required to maintain the dynamic pluripotency regulatory landscape in an accessible state.

## Introduction

Transcription factors (TFs) regulate the expression of genes through interactions with specific DNA sequences located in gene promoters and distal regulatory elements. A minority of TFs display pioneer activity, that is they have the ability to bind and induce the opening of nucleosome-occupied chromatin regions, allowing for the subsequent binding of other TFs and co-factors required for transcriptional activation (*Cirillo et al., 2002*; *Schaffner, 2015*; *Zaret and Carroll, 2011*). Pioneer TFs thereby play a central role in developmental and reprogramming cell fate decisions, which hinge on large-scale reshaping of the chromatin landscape in tissue-specific regulatory regions (*Chronis et al., 2017*; *Iwafuchi-Doi and Zaret, 2014*; *Jacobs et al., 2018*; *Pastor et al., 2018*; *Soufi et al., 2015*; *Soufi et al., 2012*; *Takaku et al., 2016*). Gain of chromatin accessibility at previously inaccessible regulatory elements has been reported to require several hours or days after pioneer TF binding (*Li et al., 2018*; *Mayran et al., 2018*). The role of pioneer TFs in maintaining the accessibility of regions that are already open has been much less studied, and little is known about pioneer activity dynamics over the cell cycle.

The OCT4 (encoded by *Pou5f1*) and SOX2 pioneer TFs (*Soufi et al., 2015*) are absolutely required for the self-renewal of embryonic stem (ES) cells (*Masui et al., 2007*; *Niwa et al., 2000*). OCT4 and SOX2 can form a heterodimer that binds to a composite motif at thousands of sites in the genome (*Boyer et al., 2005*; *Nishimoto et al., 1999*; *Yuan et al., 1995*). A recent study has shown that depletion of OCT4 for 24 hr in ES cells leads to loss of accessibility and co-factor occupancy at a large fraction of its bound enhancers involved in pluripotency maintenance (*King and Klose, 2017*). In contrast, the role of SOX2 in the regulation of ES cell chromatin accessibility has not been

**\*For correspondence:**
david.suter@epfl.ch

**Competing interests:** The authors declare that no competing interests exist.

elucidated. Thus, to which extent the pioneering activities of OCT4 and SOX2 overlap and/or depend on each other to regulate chromatin accessibility in ES cells is unclear.

Self-renewal requires the ability to progress through the cell cycle without losing cell type-specific gene expression. This is not a trivial task since chromatin accessibility of gene regulatory elements is markedly decreased during S phase and mitosis (*Festuccia et al., 2019*; *Hsiung et al., 2015*; *Oomen et al., 2019*; *Stewart-Morgan et al., 2019*). How recovery of chromatin accessibility after DNA replication and mitosis is controlled and whether it requires pioneer activity is poorly understood. The period of genome reactivation occurring at the mitosis-G1 (M-G1) transition coincides with a particularly favorable context for reprogramming by somatic cell nuclear transfer (mitosis) (*Egli et al., 2008*) and increased sensitivity to differentiation signals in human ES cells (G1 phase) (*Pauklin and Vallier, 2013*). Recent evidence also points at cell cycle stage-specific functions of OCT4 and SOX2 in cell fate regulation. OCT4 expression levels in G1 phase affect the propensity of ES cells to differentiate towards neuroectoderm and mesendoderm (*Strebinger et al., 2019*), and depletion of OCT4 at the M-G1 transition impairs pluripotency maintenance of ES cells and leads to a lower reprogramming efficiency upon overexpression in mouse embryonic fibroblasts (*Liu et al., 2017*). Depletion of SOX2 at the M-G1 transition impairs both pluripotency maintenance and SOX2-induced neuroectodermal differentiation of ES cells upon release of pluripotency signals (*Deluz et al., 2016*). Whether the particular sensitivity of M and G1 phases to the action of OCT4 and SOX2 is related to the dynamics of their pioneer activity across the cell cycle is unknown.

Here we studied the interplay of OCT4 and SOX2 in regulating chromatin accessibility of ES cells and dissected the pioneer activity of OCT4 across the cell cycle. We show that most enhancers bound by both TFs depend on only one of them to maintain their open chromatin state, and that cooperative binding of OCT4 and SOX2 is mainly mediated indirectly through changes in chromatin accessibility. Using forms of OCT4 engineered for mitotic or auxin-inducible degradation, we demonstrate the role of OCT4 in continuous maintenance of chromatin accessibility throughout the cell cycle.

## Results

### OCT4 and SOX2 regulate chromatin accessibility at mostly distinct loci

OCT4 and SOX2 bind cooperatively to thousands of genomic locations in ES cells both independently and as a heterodimer on a composite OCT4::SOX2 motif. How OCT4 and SOX2 interplay to regulate chromatin accessibility in ES cells is not known. To address this question, we decided to determine genome-wide chromatin accessibility changes upon acute loss of OCT4 or SOX2. To deplete OCT4 and SOX2 from ES cells in an inducible manner, we took advantage of the ZHBTc4 (*Niwa et al., 2000*) and 2TS22C (*Masui et al., 2007*) mouse ES cell lines, in which a Tet-Off promoter regulates the expression of *Pou5f1* (encoding OCT4) and *Sox2*, respectively (*Figure 1A*). While OCT4 is fully depleted after 24 hr of doxycycline (dox) (*Niwa et al., 2000*), SOX2 is not, likely due to its longer half-life (*Masui et al., 2007*). We determined SOX2 levels by immunofluorescence staining after 26 and 40 hr of dox treatment and found that residual SOX2 expression persisted after 26 hr but not 40 hr of dox treatment (*Figure 1—figure supplement 1A*). Importantly, despite its known role in regulating expression of OCT4 (*Dunn et al., 2014*; *Strebinger et al., 2019*), SOX2 depletion for 26 or 40 hr had only a minor impact on OCT4 levels (*Figure 1—figure supplement 1A–B*). In ZHBTc4 cells, as expected, 24 hr of dox treatment led to the complete depletion of OCT4 and only mildly affected SOX2 levels (*Figure 1—figure supplement 1C–D*). Therefore, changes in chromatin accessibility upon short-term SOX2 or OCT4 loss are unlikely to be confounded by changes in expression levels of OCT4 and SOX2, respectively.

We performed ATAC-seq in ZHBTc4 cells without dox or with dox for 24 hr, and in 2TS22C cells without dox or with dox for 26 or 40 hr. We first compared chromatin accessibility changes between ZHBTc4 cells +/- dox for 24 hr in our culture conditions (serum + 2i + LIF (S2iL), see Materials and methods) to a previous dataset acquired with ZHBTc4 cells +/- dox for 24 hr but cultured in serum + LIF (SL) (*King and Klose, 2017*). The good correlation (Pearson's R = 0.7) in chromatin accessibility changes at OCT4 binding sites between culture conditions (*Figure 1—figure supplement 1E*) prompted us to take advantage of both datasets for further analysis. We next compared changes in accessibility at SOX2 binding sites in the 2TS22C cell line treated for either 26 or 40 hr with dox,

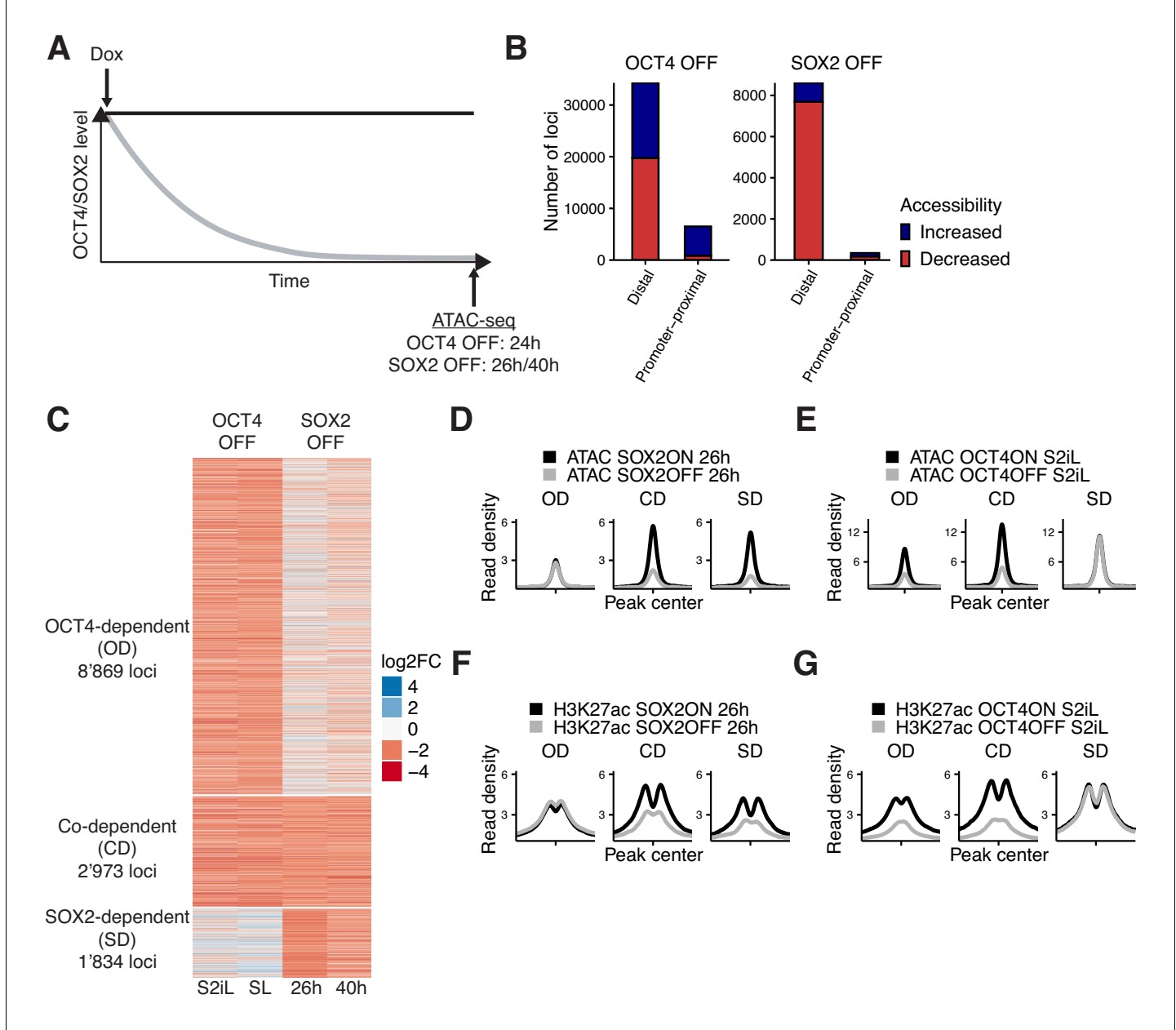

**Figure 1.** Interplay between OCT4 and SOX2 in regulating ES cell chromatin accessibility. (A) Experimental strategy to compare the effect of OCT4 and SOX2 depletion on chromatin accessibility. (B) Number of regions significantly changed in accessibility upon OCT4 (left) and SOX2 (right) depletion in distal (>1 kb from TSS) and promoter-proximal (≤1 kb from TSS) elements. (C) log2 fold-change values of accessibility between dox-treated and untreated cells upon OCT4/SOX2 depletion at OCT4/SOX2 binding sites with decreased accessibility. Loci are grouped into those significantly affected upon OCT4 depletion (OD), SOX2 depletion (SD), or depletion of either factor (CD). Each row corresponds to one individual locus, and each column to a different experimental condition. (D–E) Average RPKM-normalized ATAC-seq signal 2 kb around OD, CD, and SD loci upon SOX2 depletion (D) and OCT4 depletion (E). (F–G) Average RPKM-normalized H3K27ac ChIP-seq signal 2 kb around OD, CD, and SD loci upon SOX2 depletion (F) and OCT4 depletion (G). Statistics for (D–G) are available in *Supplementary file 1*.

The online version of this article includes the following figure supplement(s) for figure 1:

**Figure supplement 1.** Immunofluorescence analysis of OCT4 OFF and SOX2 OFF cell lines and comparison of ATAC-seq changes between culture conditions and treatment times.

**Figure supplement 2.** Heatmaps of ATAC-seq and ChIP-seq profiles in OCT4 OFF and SOX2 OFF cell lines at affected loci.

**Figure supplement 3.** Classification of OCT4/SOX2 binding sites.

which also displayed a clear correlation (Pearson's R = 0.61) (*Figure 1—figure supplement 1F*). We reasoned that the 26 hr dox dataset should be less prone to changes in accessibility due to indirect effects of prolonged SOX2 depletion than the 40 hr dox dataset, while the latter should be more sensitive to identify loci that are still accessible at low SOX2 concentrations. We thus called significantly affected loci using limma (*Ritchie et al., 2015*) (false discovery rate (FDR) < 0.05) and selected only those in which the direction of change (decrease or increase in accessibility) was the same for 26 hr vs 40 hr of dox treatment in 2TS22C cells, and likewise for SL vs S2iL in ZHBTc4 cells. In line with previous reports, loss of OCT4 led to decreased accessibility at 20'587 loci, most of which are distal regulatory elements (*Figure 1B*). Loss of SOX2 also led to decreased accessibility mainly at distal elements, but at fewer loci (7'874). We also found that loss of OCT4 led to a gain in accessibility at 20'209 loci, while 1'080 loci gained accessibility upon loss of SOX2 (*Figure 1B*). Loci that lost accessibility were highly enriched for OCT4 and SOX2 ChIP-seq binding while loci that gained accessibility were much less so (*Figure 1—figure supplement 2A–B*).

To compare the loci impacted by OCT4 vs SOX2 depletion, we next focused on all regions that were bound by OCT4 and/or SOX2 as identified from available and newly generated ChIP-seq datasets (see *Figure 1—figure supplement 2A–B* and Materials and methods) and that lost accessibility upon dox treatment. To avoid misrepresenting differences in SOX2 and OCT4 regulation that arise from differences in accessibility due to culture conditions or cell lines, we called significantly different loci (FDR < 0.05) between untreated ZHBTc4 cells cultured in SL vs S2iL conditions as well as between untreated ZHBTc4 cells and 2TS22C cells in S2iL. We then discarded all loci that displayed a large difference (FC >4) in any of those comparisons. We classified the remaining loci as OCT4-dependent (OD, 8'869 loci), SOX2-dependent (SD, 1'834 loci), and co-dependent (CD, 2'973 loci), as defined by loss of accessibility upon depletion of OCT4 only, SOX2 only, or either of them, respectively (*Figure 1—figure supplement 3A*, *Figure 1C–E*). All three groups were enriched for chromatin marks of enhancers (*Figure 1—figure supplement 3B*). We performed ChIP-seq analysis of the active enhancer mark H3K27ac (*Creyghton et al., 2010*) upon OCT4 or SOX2 loss for 24 hr and 26 hr, respectively. All groups displayed a reduction in H3K27ac, suggesting concordant maintenance of enhancer accessibility and activity by OCT4 and/or SOX2 at these loci (*Figure 1F–G*).

Surprisingly, all groups were enriched for the binding of both OCT4 and SOX2 (*Figure 2A*). 89% of SD sites overlapped with an OCT4 peak and 65% of OD sites overlapped with a SOX2 peak. Therefore, differences in the regulation of chromatin accessibility at these loci cannot simply be explained by differential DNA binding of SOX2 and OCT4. OCT4 has been shown to regulate chromatin accessibility by recruitment of the BAF chromatin remodeling complex, including the BRG1 subunit (*King and Klose, 2017*). As SOX2 also interacts with BRG1 in vivo (*Xu et al., 2018*), we asked whether SOX2 regulates chromatin accessibility through BRG1 recruitment. We performed BRG1 ChIP-seq upon SOX2 depletion and reanalyzed ChIP-seq data of BRG1 upon OCT4 depletion (*King and Klose, 2017*). We found that loss of accessibility was accompanied by loss of BRG1 in all groups (*Figure 2B–C*). We also reanalyzed ATAC-seq data from cells in which BRG1 has been depleted (*Ho et al., 2011*; *King and Klose, 2017*) and found that all groups were dependent on BRG1 to maintain their accessibility (*Figure 2—figure supplement 1A*). This suggests that OCT4 and SOX2 can regulate chromatin accessibility independently of each other even at sites that are co-occupied and through the recruitment of BRG1.

To understand which features distinguish OD, SD, and CD loci, we performed motif analysis on the underlying sequences. While both OD and CD loci were strongly enriched for the OCT4::SOX2 canonical motif and the OCT motif, SD loci were more enriched for the SOX motif (*Figure 2D–F* and *Supplementary file 2*). SD sites were also enriched for the AP-2 motif (*Figure 2—figure supplement 1B*). TFAP2C, a member of the AP-2 family, is known to regulate differentiation into trophoblast stem (TS) cells together with SOX2 (*Adachi et al., 2013*). Interestingly, when reanalyzing data from TS cells (*Adachi et al., 2013*; *Ishiuchi et al., 2019*) we found SD sites to be highly accessible and SOX2-bound compared to OD and CD loci (*Figure 2—figure supplement 1C–D*). Furthermore, SD loci were enriched near genes that increased in nascent mRNA expression upon loss of OCT4 (data from *King and Klose, 2017*) (*Figure 2—figure supplement 1E*), which by itself leads to TS cell differentiation (*Adachi et al., 2013*). In contrast, OD and CD loci were enriched near genes that decreased in nascent mRNA expression upon OCT4 depletion (*Figure 2—figure supplement 1E*). We next aimed to determine the enrichment of pluripotency-associated enhancers falling in the OD, SD, and CD groups. To this end, we checked for enrichment of the nearest gene in three gene

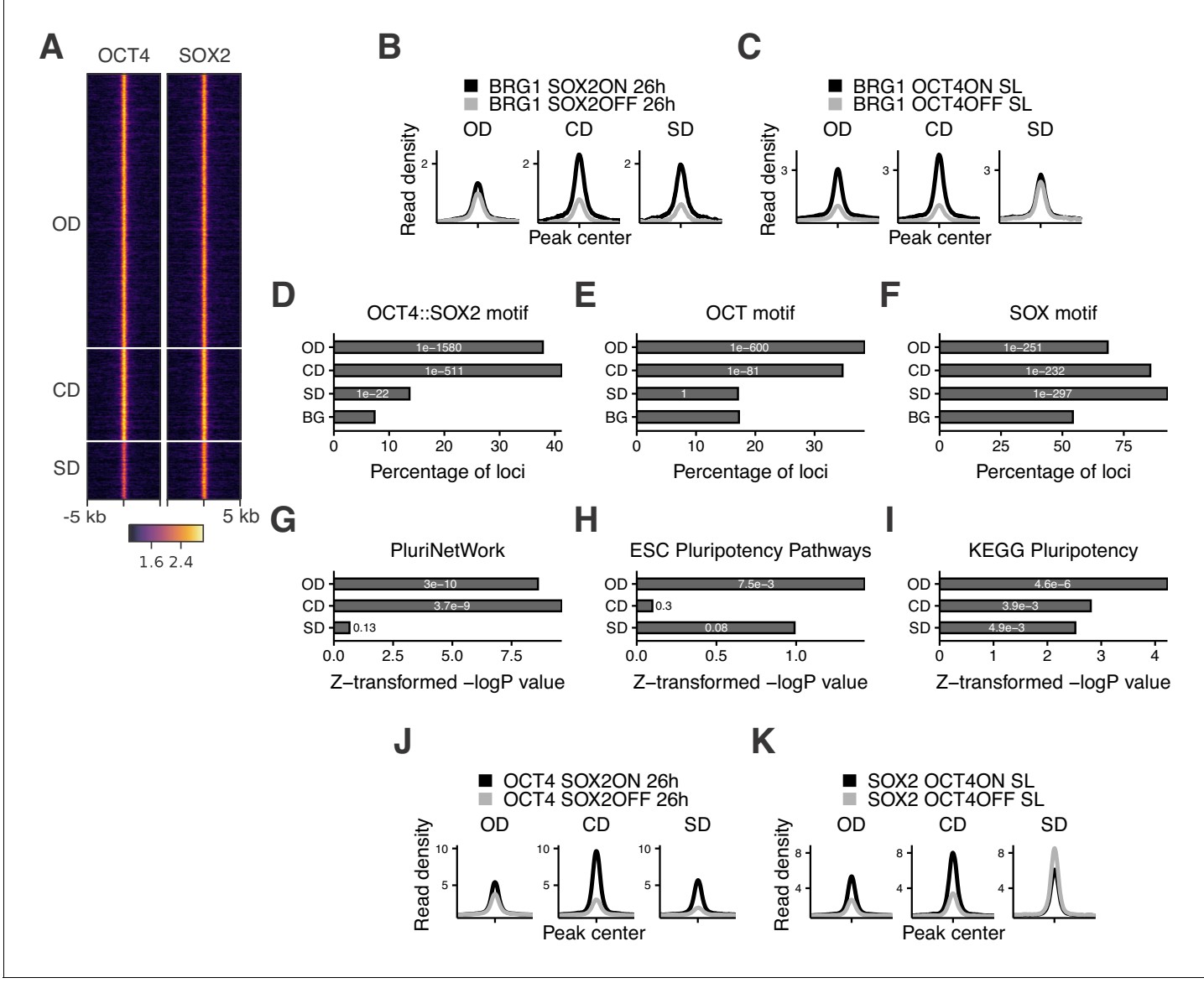

**Figure 2.** Characterization of OCT4/SOX2-dependent loci. (A) Heatmap of RPKM-normalized OCT4 and SOX2 ChIP-seq binding profiles in untreated ZHBTc4 cells 5 kb around OD, CD, and SD loci. Each row represents one individual locus. (B–C) Average RPKM-normalized BRG1 ChIP-seq signal 2 kb around OD, CD, and SD loci upon SOX2 depletion (B) and OCT4 depletion (C). (D–F) Frequency of overlap (bar) and enrichment p-values (white digits) of motifs at OD, CD, and SD loci as well as in background regions (BG) for the canonical OCT4::SOX2 motif (D), the OCT motif (E), and the SOX motif (F). (G–I) Relative enrichment values (bar) and p-values (white digits) for the closest genes in the OD, CD, and SD groups in the gene ontology sets PluriNetWork (G), ESC Pluripotency Pathways (H), and the KEGG gene set 'Signaling pathways regulating pluripotency' (I). (J–K) Average RPKM-normalized OCT4 (J) and SOX2 (K) ChIP-seq signal 2 kb around OD, CD, and SD loci upon SOX2 depletion (J) and OCT4 depletion (K). Statistics for (B–C), (J–K) are available in *Supplementary file 1*.

The online version of this article includes the following figure supplement(s) for figure 2:

**Figure supplement 1.** Additional characterization of OD, CD, and SD loci.

**Figure supplement 2.** Additional analyses of accessibility and binding changes upon SOX2 and OCT4 depletion.

ontology (GO) sets associated specifically with pluripotency. We found that these GO sets tended to be most highly enriched in genes near OD loci, although all groups were enriched in at least one of the GO sets (*Figure 2G–I*). We also analyzed the binding profiles of other pluripotency TFs (ESRRB, NANOG, KLF4, SALL4) (data from *Aksoy et al., 2014*; *Chronis et al., 2017*; *Kim et al., 2018*; *Xiong et al., 2016*) and found the highest enrichment in the CD group, although all these TFs bound to some extent to all groups of loci (*Figure 2—figure supplement 1F*). Notably, all groups were

also enriched for the 'cell differentiation' GO term (*Figure 2—figure supplement 1G*), in line with the role of OCT4 and SOX2 in ES cell differentiation. Since SOX2 was shown to require PARP1 to bind to a subset of genomic regions in ES cells (*Liu and Kraus, 2017*), we asked whether PARP1 dependence could explain the differential regulation of chromatin accessibility between these groups. We thus reanalyzed data from wt and PARP1 knockout (KO) ES cells (*Gao et al., 2009*; *Yang et al., 2004*), and found a reduction of SOX2 binding in PARP1 KO cells at all groups of loci (*Figure 2—figure supplement 1H*). Thus, PARP1 dependence cannot explain the differential regulation of chromatin accessibility between OD, CD, and SD loci. Overall, these results indicate that OCT4 and SOX2 regulate partially independent sets of pluripotency and differentiation enhancers, with OCT4 having the largest influence on chromatin accessibility of pluripotency-associated regulatory elements.

## Cooperative binding between OCT4 and SOX2 is mainly mediated indirectly through changes in chromatin accessibility

Several lines of evidence suggest that OCT4 and SOX2 exhibit cooperative DNA binding. In vitro electrophoretic mobility shift assays and fluorescence correlation spectroscopy experiments have shown that OCT4 and SOX2 display enhanced binding to the OCT4::SOX2 motif when binding together (*Mistri et al., 2015*; *Mistri et al., 2018*). While in vitro experiments reported OCT4-assisted binding on a purified nucleosomal template (*Li et al., 2019*), single-molecule imaging in live ES cells (*Chen et al., 2014*) and ChIP-seq analysis of OCT4 in the presence or absence of SOX2 in fibroblasts (*Raccaud et al., 2019*) have provided evidence that SOX2 assists OCT4 binding in vivo. However, while these experiments suggest that OCT4 and SOX2 can display direct cooperativity, the role this mechanism plays in their colocalization in the complex in vivo chromatin and nuclear environment is unclear. We reasoned that the independent regulation of chromatin accessibility by OCT4 and SOX2 at a large number of loci could result in indirect cooperativity, that is each TF could assist the binding of the other through increasing chromatin accessibility. In line with this hypothesis, it was previously shown that upon loss of OCT4, SOX2 binding loss is correlated to the loss in chromatin accessibility (*King and Klose, 2017*). However, since in vivo evidence points at a role for SOX2 in mediating cooperative OCT4 DNA-binding rather than vice versa (*Chen et al., 2014*; *Raccaud et al., 2019*), we interrogated the genome-wide binding of OCT4 upon loss of SOX2 using ChIP-seq in 2TS22C cells treated with dox for 26 hr. We found that changes in OCT4 binding were also highly correlated to changes in chromatin accessibility upon SOX2 loss (Pearson's R = 0.77) (*Figure 2—figure supplement 2A*). We next analyzed OCT4 and SOX2 binding in the presence or absence of SOX2 and OCT4, respectively, at OD, CD, and SD loci. We found that OCT4 binding was only slightly decreased at OD sites in the absence of SOX2, while SOX2 binding at SD sites was mildly increased in the absence of OCT4 (*Figure 2J–K*). These findings were also consistent when narrowing down our analysis to sites containing a canonical OCT4::SOX2 motif, although SOX2 binding did not increase at these SD sites in the absence of OCT4 (*Figure 2—figure supplement 2B–E*). The slight loss of OCT4 binding at OD sites upon only minor changes in accessibility suggests that other mechanisms such as recruitment by SOX2 may play a role in the binding of OCT4, in line with SOX2 enhancing OCT4 binding (*Figure 2J*).

Upon loss of its partner protein, OCT4 loses binding at 8'324 loci (of which 7'638 are called OCT4 peaks, representing 31% of OCT4 sites) and gains binding at 739 loci (of which 212 are called OCT4 peaks, representing 1% of OCT4 sites). Conversely, SOX2 loses binding at 6'892 loci (of which 5'302 are called SOX2 peaks, representing 29% of SOX2 sites) and gains binding at 4'136 loci (of which 983 are called SOX2 peaks, representing 5% of SOX2 sites). This indicates that the ability of OCT4 to occupy its specific binding sites is more impacted by the absence of SOX2 than vice-versa, and that SOX2 can get rerouted to new loci in the absence of OCT4. We further noticed that loci gaining accessibility upon loss of OCT4, which are enriched for differentiation terms (*Figure 2—figure supplement 1G*), also gained binding by SOX2 (see *Figure 1—figure supplement 2A* columns 6–7 bottom half) and were enriched for the SOX and AP-2 motifs (*Supplementary file 2*). 3'270 loci displayed a significant increase in both accessibility and SOX2 binding. Interestingly, these loci decreased their accessibility upon SOX2 loss (*Figure 2—figure supplement 2F*) and gained BRG1 occupancy concomitantly with OCT4 loss (*Figure 2—figure supplement 2G*), in line with SOX2 recruiting the BAF complex to promote chromatin opening. This may suggest that OCT4 sequesters SOX2 to OCT4-SOX2 sites, and upon OCT4 loss SOX2 is free to bind and increase the accessibility

of differentiation-associated regulatory elements. Overall, these results indicate that cooperative binding of OCT4 and SOX2 in ES cells is mainly mediated indirectly through changes in chromatin accessibility. However, while SOX2 enhances OCT4 binding in general, the presence of OCT4 reroutes SOX2 to pluripotency loci.

## OCT4 is required at the M-G1 transition to re-establish enhancer accessibility

Transient depletion of OCT4 or SOX2 at the M-G1 transition has been shown to hinder pluripotency maintenance (*Liu et al., 2017*; *Deluz et al., 2016*), but the underlying mechanisms are not known. This time window coincides with enhancer reopening upon chromatin decompaction, but whether pioneer factors are involved in this process is not clear. As we found OCT4 to have the broadest influence on accessibility of pluripotency-associated loci, we focused on its role in regulating chromatin accessibility at the M-G1 transition. To allow near-complete loss of OCT4 at the M-G1 transition, we used ZHBTc4 cells constitutively expressing OCT4 fused to a SNAP-tag and a Cyclin B1 degron (mitotic degron; MD) or a mutated version thereof (MD*; *Figure 3A*), which have been described previously (*Kadauke et al., 2012*). Importantly, lower than wildtype levels of OCT4 have been shown to sustain or even enhance pluripotency (*Karwacki-Neisius et al., 2013*; *Radzisheuskaya et al., 2013*). We thus reasoned that OCT4 levels need to decrease below a certain threshold to impact chromatin accessibility of pluripotency regulatory elements. Furthermore, the MD strategy strongly decreases but does not fully eliminate the target protein (*Deluz et al., 2016*; *Liu et al., 2017*). We therefore expressed MD-OCT4 and MD*-OCT4 at lower than wildtype levels from the constitutively active but relatively weak PGK promoter. After lentiviral transduction of the constructs, we stained cells with the SNAP-Cell 647-SiR dye (*Lukinavičius et al., 2013*) and sorted for a narrow window of SNAP expression to obtain the same average level of OCT4 tagged with MD and MD* across the cell cycle, as described previously (*Deluz et al., 2016*) (*Figure 3—figure supplement 1A*). We also transduced cells to express YPet-MD in a constitutive manner, which allows for discrimination between cells in early G1 (YPet-negative) and late G1 phase (YPet-positive). In combination with Hoechst staining, this enables sorting cells in early G1 (EG1), late G1 (LG1), S, and late S/G2 (SG2) phase as described previously (*Kadauke et al., 2012*) (*Figure 3—figure supplement 1B*). SNAP-MD-OCT4 levels were correlated to YPet-MD levels in flow cytometry, indicating that OCT4 levels are restored in LG1 in MD-OCT4 cells (*Figure 3—figure supplement 1C*), as shown previously (*Liu et al., 2017*). In the absence of dox, these cell lines display no substantial difference in chromatin accessibility at OCT4-regulated loci (*Figure 3—figure supplement 1D*). When grown in the presence of dox, MD*-OCT4 cells maintain a higher fraction of dome-shaped colonies, higher alkaline phosphatase activity, higher expression of pluripotency markers and lower expression of differentiation markers (*Figure 3—figure supplement 1E–G*) than MD-OCT4 cells, as also shown previously (*Liu et al., 2017*).

To test whether depletion of OCT4 at the M-G1 transition affects chromatin accessibility, we treated cells with dox for 40 hr to ensure that all cells have gone through at least one cell division expressing only MD or MD*-tagged OCT4. Note that dox-treated cells had a longer G1 phase as compared to wt ES cells, as shown before to be a consequence of lower than wt OCT4 levels (*Lee et al., 2010*). However, there was only a minor, albeit statistically significant, difference in the proportion of cells in LG1 between MD-OCT4 and MD*-OCT4 (*Figure 3—figure supplement 1H*). We sorted cells in EG1, LG1, S, and SG2 phases, performed ATAC-seq, and compared the accessibility between MD-OCT4 and MD*-OCT4 cells at each cell cycle phase (*Figure 3A*). OCT4-regulated loci that increased or decreased in accessibility upon complete OCT4 depletion (see *Figure 1B*) were also affected by transient M-G1 degradation (*Figure 3B–C*, *Figure 3—figure supplement 1I–J*). This shows that OCT4 is required at the M-G1 transition to restore chromatin accessibility and that loci gaining accessibility upon OCT4 loss are also dynamically regulated by OCT4 levels.

To characterize the different dynamic behaviors of chromatin accessibility changes across the cell cycle, we used k-means clustering on the change in accessibility between MD-OCT4 and MD*-OCT4 cells at all accessible loci displaying an OCT4 ChIP-seq peak (*Figure 3D*). Two clusters displayed decreased accessibility in EG1 and recovered their accessibility incompletely (cluster 1) or completely (cluster 2) over the cell cycle. Notably, cluster 2 loci were less affected in EG1 than cluster 1 loci, which likely explains their complete recovery. Cluster 3 loci were characterized by a minor decrease in accessibility but that persisted over the cell cycle, and cluster 4 loci were unaffected by OCT4

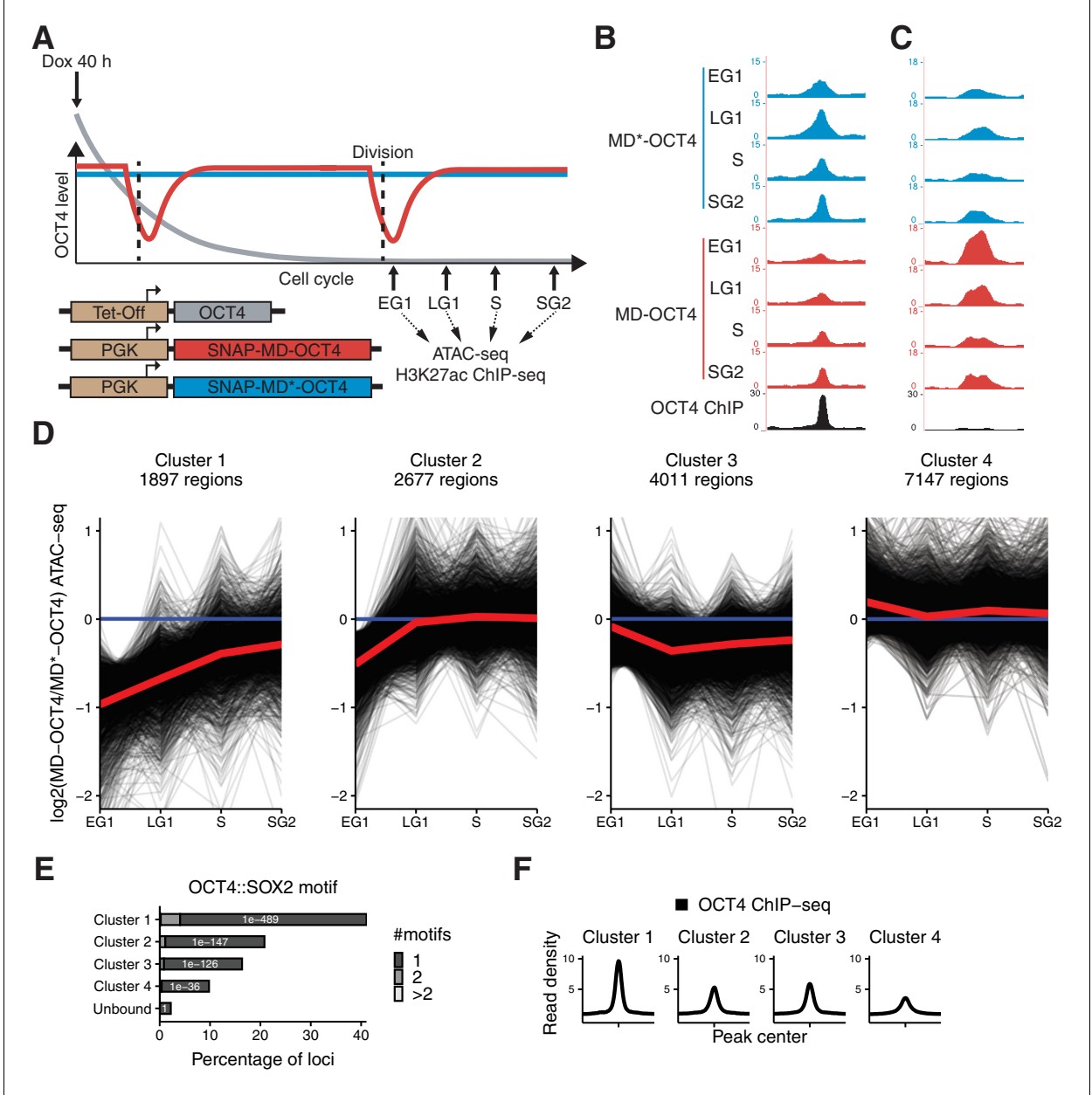

**Figure 3.** Mitotic degradation of OCT4 results in different patterns of accessibility loss. (A) Experimental strategy used to assess the impact of OCT4 depletion at the M-G1 transition. (B–C) Genome browser tracks of RPKM-normalized accessibility profiles across the cell cycle for one locus that decreases (B) at chr11:6894809–6895533 and one that increases (C) at chr9:41247953–4124841 in accessibility upon transient OCT4 depletion in M-G1. (D) log2 fold-change values of accessibility between MD-OCT4 and MD*-OCT4 (control) cells in different cell cycle phases at all accessible OCT4-bound sites, grouped into four clusters by k-means clustering (see Materials and methods). Each line represents one locus. Red line: mean. (E) Frequency of overlap (bar) and enrichment p-values (white digits) of the canonical OCT4::SOX2 motif in the four clusters as well as in unbound regions. The color shows the number of identified OCT4::SOX2 motifs per region. (F) Average RPKM-normalized OCT4 ChIP-seq signal in untreated ZHBTc4 cells 2 kb around loci in the four clusters. Statistics are available in *Supplementary file 1*. EG1: Early G1 phase; LG1: Late G1 phase; S: S phase; SG2: Late S and G2 phase.

The online version of this article includes the following figure supplement(s) for figure 3:

**Figure supplement 1.** Characterization of MD-OCT4 and MD*-OCT4 cell lines.

**Figure supplement 2.** Analyses of distance to closest gene and histone modifications in clusters.

**Figure supplement 3.** Additional analyses of clusters.

**Figure supplement 4.** Correlation between OCT4 binding and chromatin accessiblity, and analysis of results from random forest model.

loss. Cluster 1 loci were more distally located from transcription start sites (TSS) compared to the other clusters, especially cluster 4 (*Figure 3—figure supplement 2A*). Cluster 4 was also most enriched for the H3K4me3 promoter mark (*Cao et al., 2018*) (*Figure 3—figure supplement 2B*), in line with OCT4 generally not affecting accessibility at promoters (*Figure 1B* and *King and Klose, 2017*). In contrast, clusters 1–3 were most enriched for active enhancer marks (H3K4me1 and H3K27ac) (data from *Kumar et al., 2016*; *Rickels et al., 2017*) (*Figure 3—figure supplement 2B*). To test whether active histone marks also acutely change upon rapid loss of OCT4, we performed ChIP-seq for H3K27ac across the cell cycle in cells expressing MD-OCT4 or MD*-OCT4. The difference in H3K27ac across the cell cycle between the SNAP-MD-OCT4 and SNAP-MD*-OCT4 cell lines mimicked the corresponding changes in accessibility, although with smaller amplitude (*Figure 3—figure supplement 2C–F*), suggesting that this modification is also highly dynamic and sensitive to OCT4 levels.

We then further characterized these clusters by analyzing their enrichment in proximity to genes regulated by OCT4. Loci in clusters 1–3 were enriched near genes that decreased in nascent mRNA expression upon loss of OCT4, in line with these loci being involved in transcriptional regulation (*Figure 3—figure supplement 3A*). We next performed GO analysis using the KEGG database and compared the relative enrichment in the clusters (*Figure 3—figure supplement 3B*). Interestingly, cluster 1 loci were most enriched near genes annotated for pluripotency and least enriched close to genes involved in MAPK signaling, which promotes the exit from the naïve pluripotent state (*Kunath et al., 2007*). Loci in clusters 1–3 were enriched near genes annotated for PI3K-Akt signaling, which is also involved in the regulation of pluripotency (*Yu and Cui, 2016*). This suggests that all three OCT4-dependent clusters contain regulatory elements involved in pluripotency maintenance.

We analyzed the fraction of regions in the clusters overlapping previously annotated ES cell super-enhancers (SEs) and 'typical' enhancers (TEs) (*Sabari et al., 2018*; *Whyte et al., 2013*). We found these to be enriched in all clusters compared to non-OCT4 bound accessible regions, with slightly more enrichment in clusters 1 and 3 for both SEs and TEs (*Figure 3—figure supplement 3C*). This suggests that a large fraction of both SEs and TEs are affected across one cell cycle by the transient loss of OCT4 at the M-G1 transition.

As mentioned above, pluripotency was shown to be maintained at lower than wildtype OCT4 expression levels. To ask whether chromatin accessibility of the observed clusters was OCT4 level-dependent within a higher OCT4 concentration range, we interrogated chromatin accessibility in the context of physiological variations of OCT4 levels. To do so, we took advantage of ATAC-seq data we previously acquired from cells differing in their OCT4 levels by a factor of ~2, due to temporal fluctuations in their endogenous levels (*Strebinger et al., 2019*). We compared chromatin accessibility of the clusters for cells expressing high versus low endogenous levels of OCT4 and found only very minor differences in chromatin accessibility between these groups across all clusters (*Figure 3—figure supplement 3D*), consistent with the ability of moderately low OCT4 levels to fully sustain pluripotency.

To understand the reason for the differential impact of transient OCT4 depletion on chromatin accessibility, we performed motif search analysis and compared OCT4 binding at the different clusters. We found a higher enrichment for the canonical OCT4::SOX2 motif (*Figure 3E*) and a higher OCT4 occupancy (*Figure 3F*) at cluster 1 loci. Consistently, cluster 1 contained mostly OD and CD loci identified above (*Figure 3—figure supplement 3E*). As high OCT4 binding was a signature of the loci most sensitive to transient OCT4 loss, we next aimed to determine the relationship between OCT4 binding and chromatin accessibility. We compared chromatin accessibility in ZHBTc4 cells in the presence or absence of OCT4 in conditions with matched OCT4 ChIP-seq and ATAC-seq data (*King and Klose, 2017*). The OCT4 ChIP-seq signal was correlated to loss of accessibility upon OCT4 depletion (*Figure 3—figure supplement 4A*) as shown previously, but also to chromatin accessibility in untreated cells (*Figure 3—figure supplement 4B*), indicating that strong OCT4 binding sites are both highly accessible and sensitive to OCT4 levels. We also found several other motifs that were differentially enriched between the clusters, including KLF, ESRRB, NANOG, and CTCF (*Supplementary file 2*). To systematically search for differential binding in the clusters, we looked for overlap with peak regions from 5'261 ChIP-seq datasets from mouse ES cells in the cistromeDB database (*Mei et al., 2017*), of which 3'628 showed overlap with at least one region in the clusters and were used for subsequent analysis. We trained a random forest model on the data to predict which cluster a region belonged to, based on the peaks it overlapped. This model performed better

than random sampling (46.5% true positives compared to 25% expected by chance, Cohen's κ = 0.29) (*Figure 3—figure supplement 4C*), and we thus examined features used for prediction in the model to identify potential binding partners enriched in the clusters. As confirmation of the validity of the approach, the top parameters included OCT4, peaks from Dox-treated ZHBTc4 cells (e.g. SS18/SOX2/NANOG), and promoter marks (e.g. H3K4me3/H3K9ac/RPB2) (*Supplementary file 3*). We identified several factors associated with pluripotency enriched in clusters 1 and 3, including DAX1, SOX2, NANOG, and SALL4. We analyzed binding profiles of all factors described as related to pluripotency regulation in *Dunn et al. (2014)* and available in cistromeDB. We confirmed that all of these factors tended to be enriched in clusters 1 and 3, and in particular SALL4, NANOG, ESRRB, SOX2, and TBX3, which were most enriched in cluster 1 (*Figure 3—figure supplement 4D*) (data from *Aksoy et al., 2014*; *Beck et al., 2014*; *Chronis et al., 2017*; *Deluz et al., 2016*; *Han et al., 2010*; *Kim et al., 2018*; *Stevens et al., 2017*; *Xiong et al., 2016*). We also found a depletion of RAD21 (a Cohesin subunit) and CTCF in cluster 1, in line with the differential enrichment of the CTCF motif. Indeed, both RAD21 and CTCF, which are involved in the regulation of genome organization (*Phillips-Cremins et al., 2013*), were poorly bound in cluster 1 compared to the other clusters (*Figure 3—figure supplement 4E*) (data from *Cattoglio et al., 2019*). Taken together, these results reveal different classes of OCT4-bound loci that show different cell cycle accessibility dynamics upon OCT4 loss at the M-G1 transition, and that sites highly occupied by OCT4 and other pluripotency factors and lowly occupied by CTCF/Cohesin are particularly sensitive to OCT4 loss for the maintenance of their accessibility and H3K27 acetylation.

## OCT4 is required throughout the cell cycle to maintain enhancer accessibility

We next asked whether OCT4 also plays a role in maintaining enhancer accessibility in other cell cycle phases. To do so, we generated a cell line allowing drug-inducible degradation of OCT4. Briefly, we used lentiviral vectors to constitutively express the Tir1 ubiquitin ligase (allowing Auxin-inducible ubiquitination and degradation of target proteins [*Dharmasiri et al., 2005*; *Kepinski and Leyser, 2005*]) and OCT4 fused to mCherry and an Auxin-inducible degron tag (*Morawska and Ulrich, 2013*; *Nishimura et al., 2009*) (mCherry-OCT4-AID) in ZHBTc4 cells (*Figure 4A*). To verify the functionality of this fusion protein, we transduced ZHBTc4 cells with a lentiviral vector allowing expression of mCherry-OCT4-AID under the control of the constitutive EF1α promoter. After sorting for mCherry-positive cells, we plated them at low density and cultured them for one week in the presence of dox to deplete endogenous OCT4 expression, as described previously in *Strebinger et al. (2019)*. We found that these cells maintained their pluripotency, thus confirming the proper function of mCherry-OCT4-AID (*Figure 4—figure supplement 1A-B*). To ensure near-complete OCT4 depletion upon auxin treatment, we then generated a ZHBTc4 cell line in which OCT4-AID is constitutively expressed at low levels using the PGK promoter. We further expressed YPet-MD in this cell line to allow for cell sorting in different cell cycle phases, as described above. Upon addition of indole-3-acetic acid (IAA, also known as Auxin), the AID-tagged OCT4 displayed an exponential degradation profile with a half-life of 0.32 hr (*Figure 4B*). After IAA washout, OCT4 recovered to approximately half of the concentration before IAA treatment within 4.5 hr (*Figure 4C*), in line with the OCT4 protein half-life of ~4 hr (*Alber et al., 2018*).

To verify that OCT4 degradation kinetics are similar across the cell cycle, we applied IAA for 0.5 hr (partial degradation) and 2 hr (full degradation) before analyzing the mCherry signal by flow cytometry. At 2 hr of treatment, mCherry levels were similar to those of mCherry-negative cells (*Figure 4—figure supplement 1C*). We observed highly similar changes in the mCherry signal across all cell cycle phases (*Figure 4—figure supplement 1D–E*), consistent with previous reports on the cell cycle-independence of Auxin-mediated protein degradation (*Holland et al., 2012*). OCT4 recovery after IAA washout was also very similar across the cell cycle (*Figure 4—figure supplement 1F*). After addition of dox for 24 hr to remove untagged OCT4, we treated cells with IAA or not for 2 hr, sorted for different cell cycle phases, and performed ATAC-seq (*Figure 4A*). The relative magnitude of change in accessibility in the different clusters was consistent with our mitotic degradation experiment (*Figure 4D*). Remarkably, the average loss of accessibility was very similar at all cell cycle phases in clusters 1–3 (*Figure 4D*, *Figure 4—figure supplement 2A–B*). This suggests that loci in clusters 1–3, and cluster 1 in particular, are sensitive to OCT4 throughout the cell cycle and not merely at the M-G1 transition.

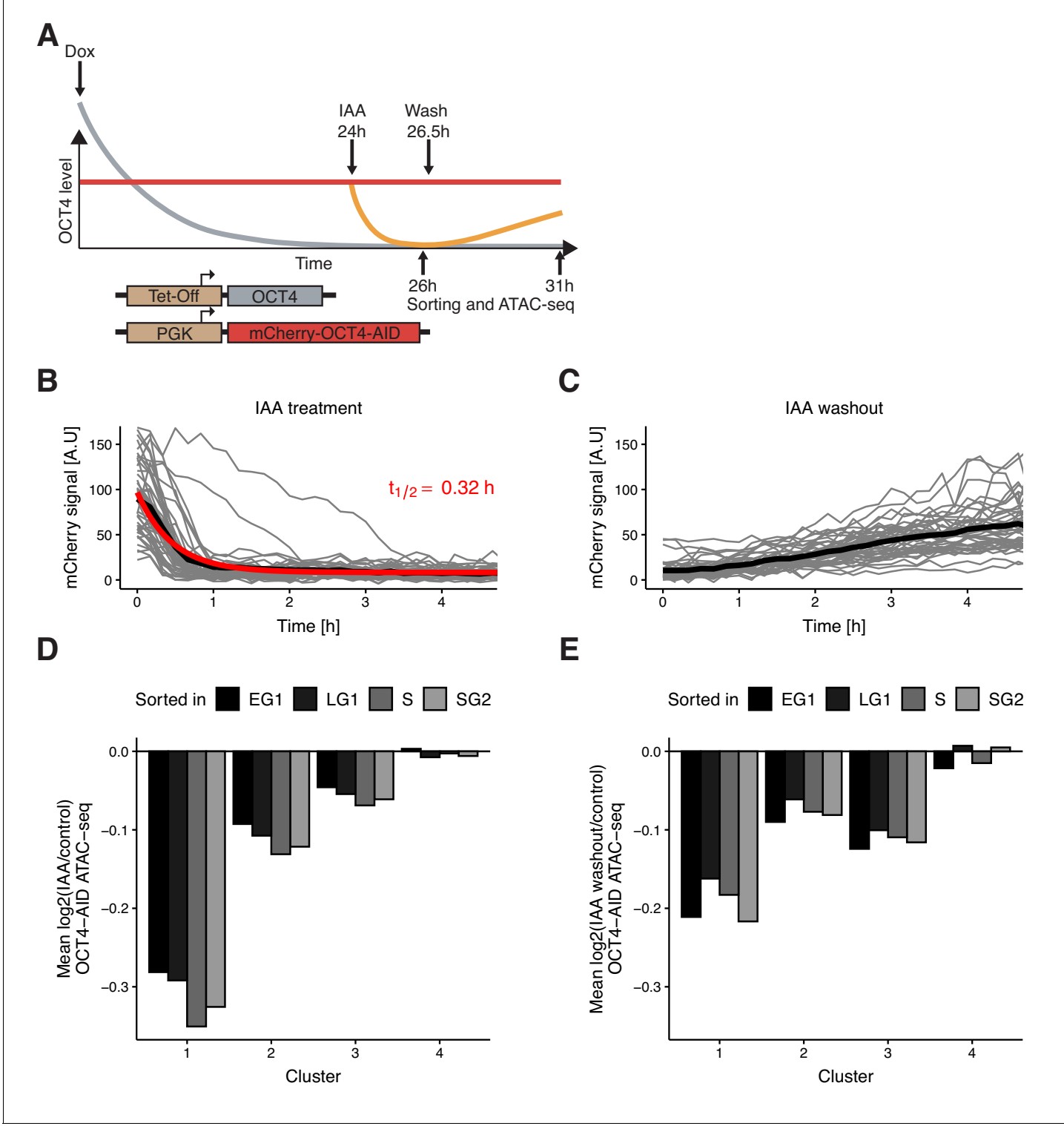

**Figure 4.** Auxin-inducible degradation reveals pioneer activity of OCT4 at different cell cycle phases. (A) Experimental strategy used to assess the impact of OCT4 depletion and recovery at different cell cycle phases. (B) Red fluorescence (mCherry) signal in mCherry-OCT4-AID cells treated with IAA at t = 0 as measured by fluorescence microscopy. Gray lines: single cell traces; Black line: population average; Red line: exponential fit. Red text: half-life value derived from the exponential fit. n = 45 cells from one replicate (C) Red fluorescence (mCherry) signal in mCherry-OCT4-AID treated with IAA for 2.5 hr and then washed out at t = 0 as measured by fluorescence microscopy. Gray lines: single cell traces; Black line: population average. n = 45 cells from one replicate (D) Average log2 fold-change values of accessibility between IAA-treated and untreated OCT4-AID cells in the four clusters from *Figure 3D* at each cell cycle phase. (E) Average log2 fold-change values of accessibility between cells first treated with IAA and then

*Figure 4 continued on next page*

*Figure 4 continued*

washed out, compared to untreated OCT4-AID cells for the four clusters from *Figure 3D* at each cell cycle phase. EG1: Early G1 phase; LG1: Late G1 phase; S: S phase; SG2: Late S and G2 phase. Statistics for (**D–E**) are available in *Supplementary file 1*.

The online version of this article includes the following source data and figure supplement(s) for figure 4:

**Source data 1.** Time-lapse microscopy source data of mCherry-OCT4-AID signal after IAA treatment (*Figure 4B*) and washout (*Figure 4C*).
**Figure supplement 1.** Characterization of mCherry-OCT4-AID cell line.
**Figure supplement 2.** Additional data on accessibility changes upon rapid IAA-mediated OCT4 depletion.

Next, we quantified the recovery of chromatin accessibility across the cell cycle. We treated OCT4-AID cells with dox for 24 hr, then with IAA or not for 2.5 hr, washed out the drug and incubated cells for 4.5 hr, sorted cells in different cell cycle phases and performed ATAC-seq (*Figure 4A*). While both cluster 1 and 2 recovered chromatin accessibility, cluster 3 loci did not (*Figure 4E*, *Figure 4—figure supplement 2C–E*), in line with their decrease of accessibility over the cell cycle upon OCT4 degradation at the M-G1 transition (see *Figure 3D*). Overall, these data show that the impact of OCT4 loss on chromatin accessibility is consistent across the cell cycle.

## Dynamic relationship between OCT4 concentration and chromatin accessibility

We next aimed to quantify the dynamics of chromatin accessibility changes in response to OCT4 loss. Since residence times of OCT4 on specific DNA sites are in the second-range (*Chen et al., 2014*; *Teves et al., 2016*; *Deluz et al., 2016*), we reasoned that if continuous OCT4 re-binding is required to maintain chromatin accessibility, changes in chromatin accessibility and OCT4 concentration should occur in a quasi-synchronized manner. To test this hypothesis, we performed a time-course experiment by treating OCT4-AID cells with IAA for 0.5 hr, 1 hr, 2 hr, 3 hr, 4 hr, 6 hr, and 10 hr, and performed ATAC-seq at each time point. We took advantage of our clusters, which showed differential response to OCT4 loss at the M-G1 transition (see *Figure 3D*), and analyzed accessibility loss at these loci over time. At all OCT4-responsive clusters (1-3), accessibility loss was near-complete after 1 hr of IAA treatment (*Figure 5A–B*), in line with accessibility being highly dynamic with OCT4 levels. At 6 and 10 hr of treatment, cluster 4 sites that were insensitive to OCT4 degradation at the M-G1 transition started to lose accessibility, suggesting a broader and potentially indirect impact of OCT4 loss on chromatin accessibility (*Figure 5A*). We thus focused on the first 4 hr of OCT4 removal to estimate the kinetics of accessibility loss. We fitted a single-component exponential function including an offset to account for the residual ATAC-seq signal after OCT4 loss. At clusters 1–3, the half-life of accessibility loss was remarkably close to the half-life of OCT4-AID upon IAA treatment, that is around 0.5 hr (*Figure 5C–E*). We were unable to fit an exponential decay to cluster 4, as expected from its OCT4-independent chromatin accessibility regulation (*Figure 5F*).

To exclude that loss of chromatin accessibility simply reflects loss of TF binding, we separately analyzed the ATAC-seq signal of subnucleosomal reads (0–100 bp) and reads from single nucleosomes (180–250 bp). Both categories of reads displayed reduced accessibility after 2 hr of IAA treatment, in line with these being bona fide changes in accessibility (*Figure 5—figure supplement 1A–E*). To test if rapid changes in chromatin accessibility impact transcription of genes regulated by these loci, we extracted RNA after 2 hr of IAA treatment and performed RT-qPCR across intron-exon and exon-exon junctions to measure pre-mRNA levels and mRNA levels, respectively. We picked several genes where the closest OCT4 peak >1 kb from TSS was a locus from clusters 1–3 and for which nascent mRNA expression was downregulated upon 24 hr OCT4 depletion according to *King and Klose (2017)*. For comparison, we also selected genes close to cluster 4 loci which were unaffected in expression after 24 hr of OCT4 depletion. All genes close to loci in clusters 1 and 2, and two out of five genes close to loci in cluster–3, showed a small decrease in pre-mRNA levels after OCT4 depletion, although only one gene (*Myc*) showed a statistically significant (p<0.05) change (*Figure 5—figure supplement 1F* and *Supplementary file 1*). In contrast, pre-mRNA levels of genes close to cluster 4 loci and mRNA (exon-exon) levels were generally unaffected, and one control gene (*Cntln*) displayed a significant increase in mRNA levels after OCT4 depletion (*Figure 5—figure supplement 1F–G* and *Supplementary file 1*). This indicates that rapid OCT4 depletion can impact transcription levels, particularly at genes close to loci regulated in accessibility by OCT4. In summary,

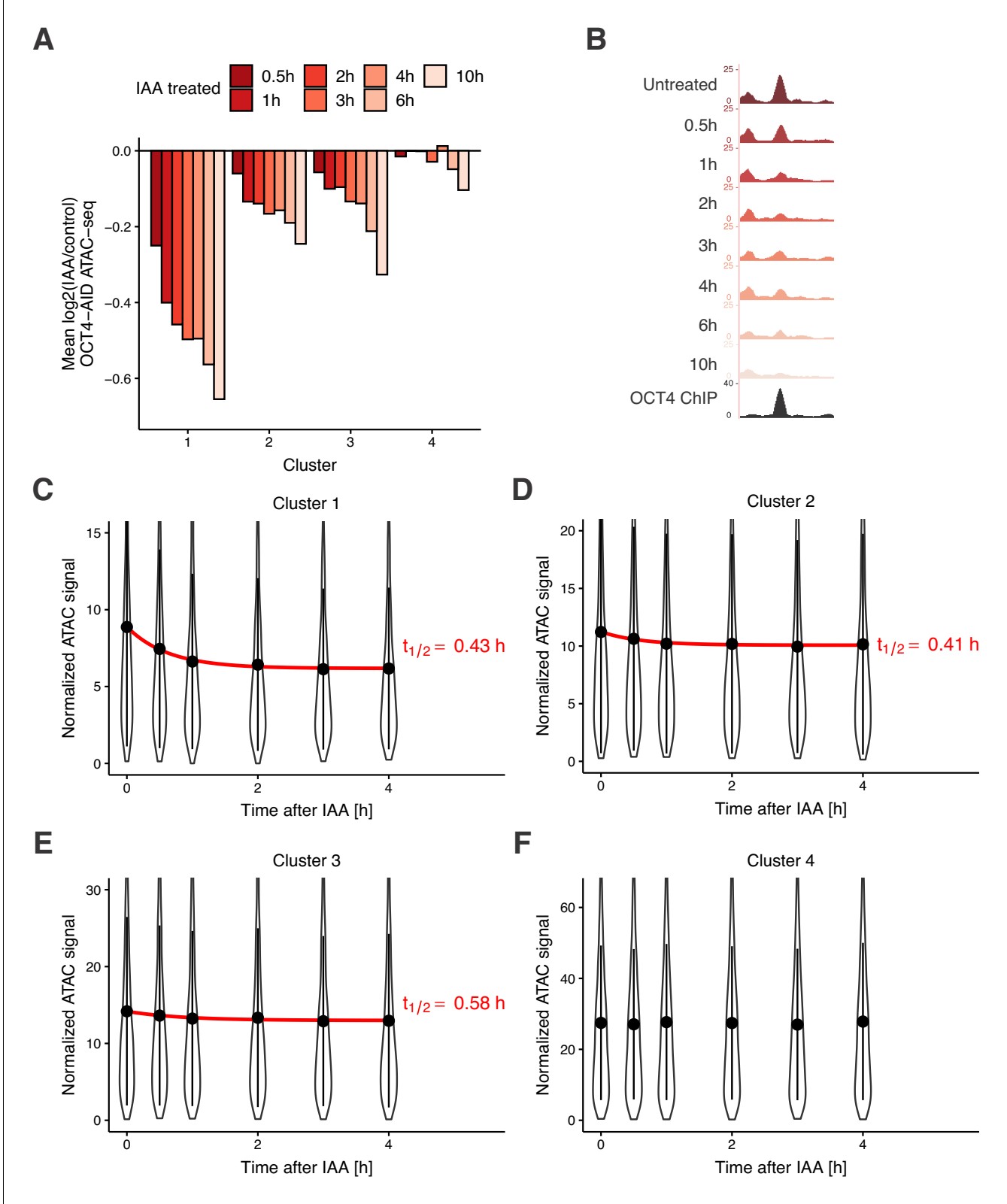

**Figure 5.** Time course analysis of chromatin accessibility changes during OCT4 degradation reveals its highly dynamic pioneer activity. (**A**) log2 fold-change values of accessibility compared to untreated cells in the four clusters from *Figure 3D* at different time points of IAA treatment. (**B**) Genome browser tracks of accessibility profiles upon treatment with IAA for different durations at a cluster 1 locus at chr3:137779908–137780687. (**C–F**) Violin

*Figure 5 continued on next page*

*Figure 5 continued*

plot of normalized ATAC-seq signal across different time points in cluster 1 (C), cluster 2 (D), cluster 3 (E), and cluster 4 (F). Dots: mean; Vertical lines: standard deviation; Red lines in C-E: exponential fit; Red text in C-E: half-life value derived from the exponential fit.

The online version of this article includes the following figure supplement(s) for figure 5:

**Figure supplement 1.** ATAC-seq data from different read sizes and RT-qPCR analysis upon rapid OCT4 depletion.

these data suggest that regulation of enhancer accessibility and activity is extremely dynamic and requires the constant presence of OCT4.

## Discussion

In this study, we dissected the roles and interplay of OCT4 and SOX2 in regulating chromatin accessibility in ES cells. To our surprise, we found a large number of enhancers that were bound by both transcription factors but for which chromatin accessibility was regulated by only one of them. In the future, it will be interesting to explore whether differences in the topology of OCT4 and SOX2 binding sites on the nucleosome surface or genomic location-dependent DNA residence times could explain these findings. While we found a larger influence on chromatin accessibility upon depletion of OCT4 than SOX2, we cannot fully exclude that this is partly caused by differences in protein half-lives or cell lines. Regions bound but not regulated by SOX2, including OD loci, could in principle also be controlled by other SOX family members such as SOX3 or SOX15 (*Corsinotti et al., 2017*; *Masui et al., 2007*). As OCT4 depletion affects accessibility already after 30 min, we can also not exclude that some of the changes in accessibility observed after long-term (24–40 hr) depletion may be due to secondary effects. Nevertheless, the differential regulation of accessibility between OCT4 and SOX2 is unlikely to be explained by these factors. Our results also show that both OCT4 and SOX2 regulate the genomic occupancy of each other mainly via regulation of chromatin accessibility. This is reminiscent of dynamic assisted loading, in which two TFs assist the loading of each other to either the same or nearby DNA binding sites (*Swinstead et al., 2016*; *Goldstein et al., 2017*).

Surprisingly, upon OCT4 loss chromatin accessibility increased at a large number of genomic sites enriched for proximity to differentiation genes, even when OCT4 was degraded for only a brief period of time at the M-G1 transition. The fact that SOX2 occupies these sites and is required to maintain their accessibility suggests that in the absence of OCT4, SOX2 is rerouted to these loci and promotes differentiation together with other partners such as TFAP2C. Therefore, the rapid action of OCT4 in early G1 phase might be required to ensure both the maintenance of chromatin accessibility at pluripotency enhancers and to silence differentiation enhancers. This is further substantiated by the fact that clusters 1 and 3 were particularly enriched for the binding of pluripotency regulatory factors and contained sites that did not fully recover upon OCT4 depletion at the M-G1 transition. Whether the previously shown association of OCT4 to mitotic chromosomes (*Deluz et al., 2016*; *Liu et al., 2017*; *Teves et al., 2016*) facilitates its action in early G1 will require further investigation.

We found that OCT4 degradation led to a rapid decrease in chromatin accessibility at all clusters of OCT4-regulated enhancers across the cell cycle with very similar kinetics, which tightly mirrored changes in OCT4 concentration and thus suggests highly dynamic regulation of chromatin accessibility by OCT4. However, the recovery of chromatin accessibility upon increase of OCT4 concentration displayed locus-dependent behavior. In contrast to clusters 1 and 2, cluster 3 loci did not recover over the time course of several hours either after M-G1 or auxin-induced degradation. While the mechanisms underlying these findings are unclear, loss across the whole cell cycle (cluster 3) or incomplete recovery (cluster 1) of chromatin accessibility may explain why OCT4 loss at the M-G1 transition results in impaired pluripotency maintenance.

Protein depletion by degron systems works by increasing protein degradation rates without affecting their synthesis rate. Therefore, they suffer from an inherent tradeoff in maximizing expression levels when the degron is inactive while minimizing residual expression level when the degron is active. Here we expressed OCT4 at relatively low levels to ensure sufficient depletion, allowing us to show that the pioneering function of OCT4 is required constantly and throughout the cell cycle to maintain enhancer accessibility. However, the low dynamic range of accessibility changes prohibits sensitive detection of specific loci that are quantitatively more or less sensitive to OCT4 loss at

different cell cycle phases. Furthermore, whether recurrent, transient loss of OCT4 outside of the M-G1 transition would also lead to pluripotency loss would have to be addressed in future studies.

Here we found that OCT4 is constantly required to maintain chromatin accessibility in self-renewing ES cells. This is reminiscent of a recent study showing that the pioneer factor Zelda is required throughout zygotic genome activation in *Drosophila* for proper gene expression (*McDaniel et al., 2019*). In contrast, in the context of pituitary lineage specification PAX7 requires 72 hr to fully open melanotrope-specific enhancers but is subsequently not required to maintain these (*Mayran et al., 2018*). It is possible that PAX7 hands over the role of maintaining accessibility to other factors, such as TPIT (*Mayran et al., 2019*), and is only required at the transition between cell fates. This indicates that pioneering activity can have different manifestations that depend heavily on other regulatory factors and chromatin features. Pluripotent stem cells may be particularly dynamic in this regard, as they need to be able to quickly rewire their gene expression programs upon receiving differentiation signals. In contrast, more differentiated cell types may have mechanisms to avoid precocious changes in gene expression upon subtle alterations in the concentration of TFs. Therefore, the high sensitivity of enhancers to the concentration or activity of pioneer TFs in ES cells could serve as a mechanism to regulate cell fate with precise temporal control.

# Materials and methods

**Key resources table**

| Reagent type (species) or resource | Designation | Source or reference | Identifiers | Additional information |
|---|---|---|---|---|
| Cell line (*Mus musculus*) | ZHBTc4 ES cells | (*Niwa et al., 2000*) | RRID: CVCL_C715 | OCT4 OFF cell line |
| Cell line (*Mus musculus*) | 2TS22C ES cells | (*Masui et al., 2007*) | RRID: CVCL_E266 | SOX2 OFF cell line |
| Antibody | Rabbit anti-BRG1 | Abcam #ab110641 | RRID: AB_10861578 | ChIP (5 µg per 10 million cells) |
| Antibody | Rabbit anti-OCT4 monoclonal | Cell Signaling Technology | RRID: AB_10547892 | ChIP (20 µl per 10 million cells) |
| Antibody | Rabbit anti-H3K27ac polyclonal | Abcam #ab4729 | RRID: AB_2118291 | ChIP 2 µg/25 µg chromatin) |
| Antibody | Mouse anti-OCT4 monoclonal | Santa Cruz #sc-5279 | RRID: AB_628051 | IF (1:500) |
| Antibody | Rabbit anti-SOX2 polyclonal | Thermo Fisher #48–1400 | RRID: AB_2533841 | IF (1:200) |
| Recombinant DNA reagent | psPAX2 | Addgene | RRID: Addgene_12260 | For lentiviral production |
| Recombinant DNA reagent | pMD2.G | Addgene | RRID: Addgene_12259 | For lentiviral production |
| Recombinant DNA reagent | pLV-PGK-YPet-MD | (*Deluz et al., 2016*) | | |
| Recombinant DNA reagent | pLV-PGK-SNAP-MD-OCT4 | This paper | | Used to generate lentiviral particles for the MD-OCT4 cell line (see *Figure 3A*). Available upon request |
| Recombinant DNA reagent | pLV-PGK-SNAP-MD*-OCT4 | This paper | | Used to generate lentiviral particles for the MD*-OCT4 cell line (see *Figure 3A*). Available upon request |

*Continued on next page*

*Continued*

| Reagent type (species) or resource | Designation | Source or reference | Identifiers | Additional information |
|---|---|---|---|---|
| Recombinant DNA reagent | pLEX-mCherry-OCT4-AID | This paper | | Used to generate lentiviral particles for the OCT4-AID cell line (see *Figure 4A*). Available upon request |
| Software, algorithm | FiJi | (*Schindelin et al., 2012*) | RRID:SCR_002285 | Version 2.0.0-rc-69/1.52 p |
| Software, algorithm | CellProfiler | (*Carpenter et al., 2006*) | RRID:SCR_007358 | Version 3.1.8 |
| Software, algorithm | STAR | (*Dobin et al., 2013*) | RRID:SCR_015899 | Version 2.6.1 c |
| Software, algorithm | Picard | Broad Institute | RRID:SCR_006525 | Version 2.8.3 |
| Software, algorithm | MACS | (*Zhang et al., 2008*) | RRID:SCR_013291 | Version 2.1.1.20160309 |
| Software, algorithm | BEDTools | (*Quinlan and Hall, 2010*) | RRID:SCR_006646 | Version 2.26.0 |
| Software, algorithm | edgeR | (*Robinson et al., 2010*) | RRID:SCR_012802 | Version 3.18.1 |
| Software, algorithm | limma | (*Ritchie et al., 2015*) | RRID:SCR_010943 | Version 3.32.10 |
| Software, algorithm | HOMER | (*Heinz et al., 2010*) | RRID:SCR_010881 | Version 4.10.4 |
| Software, algorithm | biomaRt | (*Durinck et al., 2005*) | RRID:SCR_002987 | Version 2.32.1 |
| Software, algorithm | deepTools | (*Ramírez et al., 2016*) | RRID:SCR_016366 | Version 3.2.0 |
| Software, algorithm | SAMTools | (*Li et al., 2009*) | RRID:SCR_002105 | Version 1.8 |
| Software, algorithm | GenomicRanges | (*Lawrence et al., 2013*) | RRID:SCR_000025 | Version 1.28.6 |
| Software, algorithm | RStudio | | RRID:SCR_000432 | Version 1.0.153 |
| Software, algorithm | ggplot2 | (*Wickham, 2009*) | RRID:SCR_014601 | Version 3.2.0 |

## Cell culture

Mouse ES cells were routinely cultured on cell culture-treated dishes coated with 0.1% gelatin (Sigma #G9391-100G) using the following culture medium: GMEM (Sigma #G5154-500ML) containing 10% ES-cell qualified fetal bovine serum (Gibco #16141–079), nonessential amino acids (Gibco #11140–050), 2 mM L-glutamine (Gibco #25030–024), sodium pyruvate (Sigma #S8636-100ML), 100 µM 2-mercaptoethanol (Sigma #63689–25 ML-F), penicillin and streptomycin (BioConcept #4–01 F00-H), in-house produced leukemia inhibitory factor (LIF), CHIR99021 (Merck #361559–5 MG) at 3 µM and PD184352 (Sigma #PZ0181-25MG) at 0.8 µM. Cells were passaged by trypsinization (Sigma #T4049-100ML) every two to three days. All cell lines used in this study were tested by Eurofins Genomics at the end of the study (August 2019) and found to be Mycoplasma-negative.

## Lentiviral vector production

Lentiviral vectors were produced by transfection of HEK 293 T cells with the envelope (psPAX2, Addgene #12260), packaging (pMD2.G, Addgene #12259) (*Dull et al., 1998*), and lentiviral construct of interest using Calcium Phosphate transfection, as described previously (*Suter et al., 2006*). Viral vectors were concentrated 120-fold by ultracentrifugation at 20'000 rpm for 90 min at 4°C. 50'000 cells in 1 ml of medium in a 24-well plate were transduced with 50 µl of concentrated lentiviral vector particles to generate stable cell lines.

Cloning of overexpression constructs pLV-PGK-YPet-MD was derived from pLVTRE3G-SOX2-YPet-MD (*Deluz et al., 2016*) by amplification of YPet-MD and restriction cloning into pLV-rtTA3G-IRESBsd using AgeI and SalI. pLV-PGK-SNAP-MD-OCT4 and pLV-PGK-SNAP-MD*-OCT4 were derived by amplification of MD or MD* from SNAP-MD-SOX2 (Addgene #115687) and SNAP-MD*-SOX2 (Addgene #115688) (*Deluz et al., 2016*) and restriction cloning into pLVTRE3G-OCT4-YPet (*Deluz et al., 2016*) using SalI and XbaI. SNAP-MD-OCT4 and SNAP-MD*-OCT4 were further amplified and cloned by restriction cloning into pLV-rtTA3G-IRESBsd (*Deluz et al., 2016*) using AgeI and SalI. pLV-pCAGGS-Tir1-V5 was derived by amplification of pCAGGS-Tir1-V5 from pEN395 (Addgene #92141) (*Nora et al., 2017*) and In-fusion cloning into pLV-PGK-SOX2-SNAP-IRES-Hygro

(*Strebinger et al., 2019*) digested using XhoI and XbaI restriction enzymes. pLEX-mCherry-OCT4-AID was derived by amplification of OCT4 from pLV-PGK-SNAP-MD-OCT4, AID 71–114 from pEN244 (Addgene #92140) (*Nora et al., 2017*), and mCherry from pLV-TRE3G-mCherry-PGK-Puro (Suter lab). mCherry and OCT4 were ligated using an XmaI restriction site and mCherry-OCT4 was cloned into the pLEX_305-C-dTAG backbone (Addgene #91798) (*Nabet et al., 2018*) using AgeI and NheI restriction sites. OCT4 was amplified and ligated to amplified AID 71–114 using a KpnI restriction site. The OCT4-AID fragment was cloned into the pLEX-mCherry-OCT4 vector using XmaI and MluI restriction sites. Note that this construct acquired a G161T mutation in the coding DNA sequence of OCT4, which gives rise to a G54V mutation in the OCT4 peptide sequence of mCherry-OCT4-AID. However, this is inconsequential since this construct is functional in maintaining pluripotency (*Figure 4—figure supplement 1A–B*).

## Generation of stable cell lines

To generate MD-OCT4 cell lines, ZHBTc4 cells were transduced with SNAP-MD-OCT4 and SNAP-MD*-OCT4 lentiviral vector particles and sorted to display near-identical average SNAP levels (*Figure 3—figure supplement 1A*), subsequently transduced with PGK-YPet-MD lentiviral vector particles, and sorted to display near-identical average YPet-MD levels. To generate the OCT4-AID cell line, ZHBTc4 cells were transduced with pLV-pCAGGS-Tir1-V5 and pLEX-mCherry-OCT4-AID packaged lentiviral vectors (i.e Tir1-V5 and mCherry-OCT4-AID virus, respectively) and were selected with 2 µg/ml Puromycin (Thermo Fisher Scientific #A1113803) for 10 days. Subsequently, mCherry positive cells were sorted and transduced with PGK-YPet-MD lentiviral particles and sorted for YPet positive cells. Cells that displayed IAA-dependent degradation were selected by sorting a narrow window of mCherry-positive cells, followed by treatment with 500 nM IAA (Sigma #I5148-2G) for 1 hr, and sorting for mCherry-negative cells. All cell lines were maintained in medium without dox (Sigma #D3447-500MG) or IAA prior to experiments.

## Treatment conditions

Cells were seeded at a concentration of 9'000–18'000 cells per cm$^2$ one day before the start of treatment. ZHBTc4 YPet-MD SNAP-MD-OCT4 and SNAP-MD*-OCT4 were treated with 1 mg/ml dox for 40 hr prior to cell sorting. ZHBTc4 YPet-MD TIR1-V5 mCherry-OCT4-AID cells were treated with 1 mg/ml dox for 24 hr before adding IAA. Dox was maintained throughout the experiment. Cells were treated with 500 nM IAA (or not for control) for 2 hr or treated with 500 nM IAA (or not for control) for 2.5 hr, washed five times with PBS with 2 min incubation, and placed back in medium containing 1 mg/ml dox for 4.5 hr. For the time course experiment, OCT4-AID cells were seeded in different wells of a 24-well plate and treated with dox for 24 hr before adding IAA. Dox was maintained throughout the experiment. IAA was added at different time points (with one well left untreated) prior to cell collection. All wells were collected at the same time and subjected to ATAC-seq as described below.

## Cell cycle phase sorting

Cells were trypsinized, resuspended in culture medium with 50 mM Hoechst33342 (Thermo Fisher Scientific #H3570), and incubated for 15 min at 37°C. Cells were then spun down, resuspended in cold PBS with 1% FBS, and sorted according to their YPet-MD and Hoechst profile (*Figure 3—figure supplement 1B*). Cells were sorted at 4°C into 1.5 ml Eppendorf tubes or 15 ml Falcon tubes containing a small amount of PBS with 1% FBS. Sorting for SNAP levels was done on a MoFlo Astrios (Beckman Coultier). All other sorting was done on a FACSAria II or a FACSAria Fusion (BD Biosciences).

## ATAC-seq

All ATAC-seq experiments were performed in biological duplicates except for IAA-treated mCherry-OCT4-AID cells sorted in EG1, LG1, and S phase where three replicates were performed. 50'000 cells were collected either directly after trypsinization or after sorting as described above and subjected to ATAC-seq as described previously (*Buenrostro et al., 2013*). All centrifugation steps were done at 800 g at 4°C. Briefly, cells were centrifuged for 5 min and washed with cold PBS, then centrifuged for 5 min and resuspended in cold lysis buffer (10 mM Tris-HCl pH 7.4, 10 mM NaCl, 3 mM

MgCl2, 0.1% NP-40), and centrifuged for 10 min. Subsequently, nuclei were resuspended in a solution of 0.5 mM Tn5 (in-house preparation according to *Chen et al., 2017*) in TAPS-DMF buffer (10 mM TAPS-NaOH, 5 mM Mgcl2, 10% DMF) and incubated for 30 min at 37°C. DNA was immediately purified using column purification (Zymo #D4004) and eluted in 10 ml nuclease-free H2O. Transposed DNA was amplified in a solution containing 1X NEBNextHigh Fidelity PCR Master Mix (NEB #M0541L), 0.5 µM of Ad1_noMX universal primer, 0.5 µM of Ad2.x indexing primer and 0.6x SYBR Green I (Thermo Fisher Scientific #S7585) using 72°C for 5 min, 98°C for 30 s, and 5 cycles of 98°C for 1 s, 63°C for 30 s, and 72°C for 60 s. 10 µl of amplified DNA was analyzed by qPCR to determine the total number of cycles to avoid amplification saturation and accordingly amplified with additional 3–7 cycles of 98°C for 10 s, 63°C for 30 s, and 72°C for 60 s. DNA was purified using column purification (Zymo #D4004) and size-selected by taking the unbound fraction of 0.55X AMPure XP beads (Beckman Coultier #A63880) followed by the bound fraction of 1.2X AMPure XP beads. Libraries were sequenced on an Illumina NextSeq 500 using 75 nucleotide read-length paired-end sequencing.

## ChIP-seq

All ChIP-seq experiments were performed in biological duplicates. Roughly 10 million cells per sample were collected after trypsinization and fixed with 2 mM disuccinimidyl glutarate (Thermo Fisher Scientific #20593) in PBS for 50 min at room temperature, spun down at 600 g for 5 min and washed once with PBS. Cells were then treated with 1% formaldehyde (Axon Lab #A0877,0500) for 10 min at room temperature and quenched with 200 mM Tris-HCl pH 8.0 for 10 min, washed with PBS and spun down. For ZHBTc4 YPet-MD SNAP-MD-OCT4 and SNAP-MD*-OCT4 cells, cells were subsequently resuspended in cold PBS with 1% FBS and at least 500'000 cells per cell cycle phase were sorted (see Cell cycle phase sorting section above). Fixed cell pellets were kept on ice and resuspended in LB1 (50 mM HEPES-KOH pH 7.4, 140 mM NaCl, 1 mM EDTA, 0.5 mM EGTA, 10% Glycerol, 0.5% NP-40, 0.25% Triton X-100), incubated 10 min at 4°C, spun down at 1700 g, and resuspended in LB1 a second time, spun down and resuspended in LB2 (10 mM Tris-HCl pH 8.0, 200 mM NaCl, 1 mM EDTA, 0.5 mM EGTA), incubated for 10 min at 4°C, spun down and washed without disturbing the pellet three times with SDS shearing buffer (10 mM Tris-HCl pH 8.0, 1 mM EDTA, 0.15% SDS) and finally resuspended in SDS shearing buffer. All buffers contained Protease Inhibtor Cocktail (Sigma #P8340-1ML) at 1:100 dilution. Chromatin was sonicated for 20 min at 5% duty cycle, 140 W, 200 cycles on a Covaris E220 focused ultrasonicator. Sonicated chromatin was equilibrated to 1% Triton X-100 and 150 mM NaCl and incubated with each antibody overnight at 4°C. Antibodies used were anti-BRG1 (Abcam #ab110641) at 5 µg per 10 million cells, anti-OCT4 (Cell Signaling Technology #5677S) at 20 µl per 10 million cells, and anti-H3K27ac (Abcam #ab4729) at 2 µg/25 µg chromatin. Protein G Dynabeads (Thermo Fisher Scientific #10003D) were blocked with 5 mg/ml BSA in PBS, added to chromatin, and incubated at 4°C for 3 hr. Beads were washed twice with Low Salt wash buffer (10 mM Tris-HCl pH 8.0, 150 mM NaCl, 1 mM EDTA, 1% Triton X-100, 0.15% SDS, 1 mM PMSF), once with High Salt wash buffer (10 mM Tris-HCl pH 8.0, 500 mM NaCl, 1 mM EDTA, 1% Triton X-100, 0.15% SDS, 1 mM PMSF), once with LiCl wash buffer (10 mM Tris-HCl pH 8.0, 1 mM EDTA, 0.5 mM EGTA, 250 mM LiCl, 1% NP40, 1% sodium deoxycholate, 1 mM PMSF), and finally with 1X TE (10 mM Tris pH 8.0, 1 mM EDTA) before being resuspended in ChIP Elution buffer (10 mM Tris pH 8.0, 1 mM EDTA, 1% SDS, 150 mM NaCl) with 400 ng/µl Proteinase K (Qiagen #19131) and reverse-crosslinked overnight at 65°C. DNA was purified using a MinElute PCR purification kit (Qiagen #28004) and libraries were prepared with the NEBNext Ultra II DNA Library Prep Kit (NEB #E7645S). Libraries were sequenced on an Illumina NextSeq 500 using 75-nucleotide read length paired-end sequencing.

## Pluripotency assays and RT-qPCR

ZHBTc4 YPET-MD cells expressing SNAP-MD-OCT4 or SNAP-MD*-OCT4 or ZHBTc4 cells expressing mCherry-OCT4-AID from the EF1α promoter were plated at 400 cells per well in a 6-well plate with ES cell medium (see above) with 0 or 1 µg/ml dox and medium was refreshed every other day. At day 7, flat and dome-shaped colonies were scored according to morphology and counted (SNAP-MD-OCT4 and SNAP-MD*-OCT4) or total colonies were counted (mCherry-OCT4-AID) followed by alkaline phosphatase staining (Sigma #86R-1KT). For RT-qPCR experiments, cells were collected at day 7 (*Figure 3—figure supplement 1G*) or after 26 hr of dox treatment and 2 hr with or without

IAA treatment (*Figure 5—figure supplement 1F–G*) and RNA was extracted using GenElute Mammalian Total RNA MiniPrep Kit (Sigma #RTN350). cDNA was synthesized using oligodT primers (*Figure 3—figure supplement 1G*) or random hexamers (*Figure 5—figure supplement 1F–G*) using SuperScript II Reverse Transcriptase (Thermo Fisher Scientific #18064014). qPCR was performed on a 7900HT (Applied Biosystems). The $2^{(-Ct)}$ value of each primer pair was normalized to that of Rps9 in the same sample. Primers are listed in *Supplementary file 4*.

## Immunofluorescence microscopy

2TS22C and ZHBTc4 cells were plated in a 96-well plate coated for 1 hr at 37°C with 1:25 diluted StemAdhere (Primorigen Biosciences #S2071-500UG), treated with 1 mg/ml dox for different durations and fixed with 2% formaldehyde for 30 min at room temperature, washed with PBS, permeabilized with PBS with 5% FBS and 0.5% Triton X-100 for 30 min at room temperature, and incubated with the primary antibody, anti-OCT4 C-10 (Santa Cruz #sc-5279) at 1:500 dilution and anti-SOX2 (ThermoFisher #48–1400) at 1:200 dilution, in PBS with 5% FBS and 0.1% Triton X-100 at 4°C overnight. After washing with PBS, cells were incubated with the secondary antibody, anti-Mouse IgG AF488 (Thermo Fisher Scientific #A28175) and anti-Rabbit IgG AF647 (Thermo Fisher Scientific #A27040) at 1:1000 dilution, in PBS with 5% FBS and 0.1% Triton X-100 for 60 min at room temperature, with 2 ng/ml DAPI (Thermo Fisher Scientific #62248) for 10 min at room temperature and subsequently washed and imaged on an IN Cell Analyzer 2200 (GE Healthcare). Images were background-subtracted using FiJi (*Schindelin et al., 2012*) with a rolling ball radius of 50 pixels and analyzed using CellProfiler (*Carpenter et al., 2006*). Nuclei were identified using the Watershed module on the DAPI channel, objects that were too large or too small were discarded, and the mean intensity in the OCT4 and SOX2 channels was measured within the identified nuclei.

## Time-lapse microscopy

ZHBTc4 YPet-MD TIR1-V5 mCherry-OCT4-AID cells were plated in a 96-well plate coated for 1 hr at 37°C with 1:25 diluted StemAdhere (Primorigen Biosciences #S2071-500UG) in Phenol Red-free medium and imaged on an IN Cell Analyzer 2200 (GE Healthcare) using the TexasRed and Brightfield channels. Cells were treated with IAA or washed just prior to imaging. Images were background-subtracted using FiJi (*Schindelin et al., 2012*) with a rolling ball radius of 50 pixels and nuclei were tracked manually over time using the Manual Tracking plugin in FiJi in the Brightfield channel. The mean signal in 10 pixels around the tracked spot were measured in the TexasRed channel and the mean background signal at an equivalent sized spot free from cells (background) was subtracted at each time point.

## Data analysis for ATAC-seq and ChIP-seq

All sequencing libraries were aligned to the mm10 *Mus musculus* genome (GRCm38 release 87) with STAR 2.6.1c (*Dobin et al., 2013*) and duplicate reads were removed using Picard (Broad Institute). Reads not mapping to chromosomes 1–19, X, or Y were removed. Peaks were called with MACS 2.1.1.20160309 (*Zhang et al., 2008*) with settings '-f BAMPE -g mm'. For comparative analysis of ZHBTc4 and 2TS22C cells, all peaks from ZHBTc4 and 2TS22C ATAC-seq experiments were merged with BEDTools (*Quinlan and Hall, 2010*). For MD-OCT4 and OCT4-AID analyses, peaks from all ATAC-seq experiments of dox-treated cells in the corresponding cell lines were merged. For MD-OCT4 H3K27ac analysis, peak coordinates were expanded by 500 bp on both sides to account for the enrichment profile of H3K27ac. All peaks larger than 5 kb, overlapping peaks called in Input (no immunoprecipitation) samples from ES cells in S2iL (GSE89599) or SL (GSE87822), or overlapping blacklisted peaks (*ENCODE Project Consortium, 2012*) were removed. The HOMER2 (*Heinz et al., 2010*) functions makeTagDirectory and annotatePeaks.pl with settings '-noadj -len 0 -size given' were used for read counting and count tables were loaded into RStudio. TMM Normalization was done with edgeR (*Robinson et al., 2010*) and analysis of differentially accessible regions was done with limma (*Ritchie et al., 2015*). Contrasts were designed as ~0+Condition+Replicate, where Condition specifies the cell line and treatment and Replicate the date of the experiment, to take into account the paired nature of the experiments. For comparing unpaired experiments, that is untreated ZHBTc4 vs 2TS22C cell lines or untreated ZHBTc4 in SL versus S2iL,~0+Condition was used. For *Figure 3—figure supplement 4A–B*, the mean of the TMM-normalized reads in the ChIP-

seq and ATAC-seq replicates was divided by the nucleotide length of each region. For *Figure 5C–F* and *Figure 5—figure supplement 1A–D*, the mean of the TMM-normalized reads in the replicates was used. Replicate bam files were merged using SAMTools (*Li et al., 2009*) and converted to big-Wig files using the deepTools 3.1.3 (*Ramírez et al., 2016*) function bamCoverage with settings '—normalizeUsing RPKM'. SAMTools and awk were used on merged bam files to spit reads into size classes. Average lineplots were generated using deepTools computeMatrix (with setting 'reference-point') and custom R code. Heatmaps were generated using the deepTools function plotHeatmap. Genome tracks were made in the UCSC genome browser (*Kent et al., 2002*). Plots were generated using ggplot2 (*Wickham, 2009*). Overlap between genomic regions was determined using GenomicRanges (*Lawrence et al., 2013*). Heatmaps of fold-changes were generated using ComplexHeatmap in R (*Gu et al., 2016*). Color schemes were taken from colorbrewer2.org and https://rpubs.com/Koundy/71792.

## Motif analysis and gene ontology enrichment

The HOMER2 function findMotifsGenome.pl was used with the setting '-size given' for motif searching. The most frequent known motif in target regions of a given class of known motifs (i.e. different versions of SOX and OCT motifs) was used. Background was calculated as the mean of HOMER-estimated background frequency in all groups/clusters. The HOMER2 function scanMotifGenomeWide.pl was used to determine the number of OCT4::SOX2 motifs per region (*Figure 3E*). For GO enrichment analysis, the closest Entrez gene entry TSS to each region was used, enrichment was calculated using the HOMER2 function findGO.pl with setting 'mouse', and -logP values within each category (e.g. KEGG) were Z-transformed. Gene names were converted between assemblies using biomaRt (*Durinck et al., 2005*). Only GO categories with a Z-score >3 are shown in the heatmap in *Figure 3—figure supplement 3B*, which was generated using pheatmap in R.

## Random forest model

The model was generated as described previously (*Strebinger et al., 2019*). Briefly, all peak files in the Mouse_Factor and Mouse_Histone categories in cistromeDB (*Mei et al., 2017*) were batch downloaded (http://cistrome.org/db) and for all peak files where the cell type was 'Embryonic Stem Cell' overlap with our cluster regions was determined using the BEDTools function intersect. Regions were randomly sampled from clusters 2–4 to contain the same number of regions as cluster 1 (n = 1'897). These were split into training (80%, n = 6'070) and test (20%, n = 1'518) regions. The function randomForest in R was used with the settings 'formula = Cluster ~ ., mtry = 4, ntree = 2001' on the training regions. The predict function in R was used with setting 'type = "response"' to use the model for predicting the clusters of the regions in the test data.

## Published datasets

Published data (see *Supplementary file 5*) were aligned and processed as described above. Processed bigWig files (see *Supplementary file 5*) were downloaded from GEO (*Edgar et al., 2002*) or cistromeDB (*Mei et al., 2017*). When necessary, peak files were converted to mm9 using liftOver (*Hinrichs et al., 2006*). OCT4 and SOX2 ChIP-seq peaks were derived from newly generated (2TS22C OCT4) and published (ZHBTc4 OCT4 and SOX2) (*King and Klose, 2017*) datasets as described above as well as from processed SOX2 ChIP-seq peaks from asynchronous E14 cells (GSE89599) (*Deluz et al., 2016*) and merged with BEDTools. Super-enhancers and typical enhancers were taken from *Sabari et al. (2018)* and converted to mm10 using liftOver. ChromHMM tracks from mouse ES cells were downloaded from https://github.com/guifengwei/ChromHMM_mESC_mm10 (*Pintacuda et al., 2017*). ATAC-seq data from OCT4 high and OCT4 low sorted cells were taken from a previous study (GSE126554) (*Strebinger et al., 2019*) and processed as described above, merging SHOH and SLOH samples into OCT4 high and SLOL and SHOL samples into OCT4 low.

## K-means clustering

Clusters in *Figure 3D* were generated using the R function pheatmap with settings 'clustering_distance_rows = "euclidean", kmeans_k = 4' on a matrix containing the log2 fold-change values in

accessibility between MD-OCT4 and MD*-OCT4 at each cell cycle phase (columns) at each OCT4-bound locus (rows). Clusters were ordered according to the lowest mean log2 fold-change in EG1.

## Exponential curve fitting

Exponential decays were fitted using the R function nls with the formula $y \sim a*e^{(-b*x)+c}$ where a, b, and c are constants. Half-life values were derived as log(2)/b.

## Statistical analysis

Distributions were tested for non-normality using the Shapiro-Wilk test and if p<0.05 for any of the samples a non-parametric (Mann-Whitney U, or Wilcoxon signed rank test if paired) test was used, otherwise a t-test was used. These were performed using two-tailed distributions and with unequal variance. In cases where comparison experiments were matched (e.g. treatment and control handled at the same time), paired testing was performed, except for non-parametric tests with n < 5 where the Wilcoxon signed rank test cannot give precise p-values. For average lineplots (e.g. *Figure 1D*), distributions of RPKM-normalized reads derived from deepTools matrices on merged bigWig files were compared at x = 0 (Peak center) except for H3K27ac ChIP-seq (*Figure 1F–G*) where comparisons were done at x = +/- 400 (*Supplementary file 1*). Fisher's exact tests were performed on a 2 × 2 matrix with the number of regions in the category of interest and the total number of regions in the two comparison groups. Correlation p-values were generated using the stat_cor function in the ggpubr package in R.

## Acknowledgements

This work was supported by the Swiss National Science Foundation (grants #PP00P3_179068 and PP00P3_17205 to DMS). ACAMF was supported by a Marie Curie Intra European Fellowship within the 7th European Community Framework Programme. This work was further supported by AgingX (SystemsX.ch) and SNF (310030_182655). We thank Bastien Mangeat, Elisa Cora, Paolo Ferrari, and Lionel Ponsonnet from the Gene Expression Core Facility for high-throughput sequencing, Miguel Garcia, Loïc Tauzin, Valérie Glutz, and André Mozes from the Flow Cytometry Core Facility for cell sorting, Olivier Burri and Romain Guiet from the Bioimaging and Optics Platform for assistance with cell tracking, the staff at Vital-IT and SCITAS for cluster computing, and Armelle Tollenaere for critical reading of the manuscript.

## Additional information

### Funding

| Funder | Grant reference number | Author |
|---|---|---|
| H2020 Marie Skłodowska-Curie Actions | Marie Curie Intra Europen Fellowship | Antonio CA Meireles-Filho |
| SystemsX | AgingX | Bart Deplancke |
| Swiss National Science Foundation | PP00P3_179068 | David M Suter |
| Swiss National Science Foundation | 310030_182655 | Bart Deplancke |

The funders had no role in study design, data collection and interpretation, or the decision to submit the work for publication.

### Author contributions

Elias T Friman, Conceptualization, Formal analysis, Investigation, Visualization, Methodology, Writing—original draft, Project administration, Writing—review and editing; Cédric Deluz, Investigation, Methodology; Antonio CA Meireles-Filho, Investigation; Subashika Govindan, Methodology; Vincent Gardeux, Formal analysis, Writing—review and editing; Bart Deplancke, Resources, Writing—review

and editing; David M Suter, Conceptualization, Resources, Supervision, Funding acquisition, Writing—original draft, Project administration, Writing—review and editing

### Author ORCIDs
Elias T Friman (iD) https://orcid.org/0000-0001-9944-6560
Vincent Gardeux (iD) http://orcid.org/0000-0001-8954-2161
Bart Deplancke (iD) http://orcid.org/0000-0001-9935-843X
David M Suter (iD) https://orcid.org/0000-0001-5644-4899

### Decision letter and Author response
Decision letter https://doi.org/10.7554/eLife.50087.sa1
Author response https://doi.org/10.7554/eLife.50087.sa2

## Additional files
### Supplementary files
• Supplementary file 1. Additional statistics.This file contains p-values and fold change values for all relevant comparisons in *Figure 1D–G*, *Figure 2B,C,J,K*, *Figure 2—figure supplement 1A,C,D,F,H*, *Figure 2—figure supplement 2B–G*, *Figure 3F*, *Figure 3—figure supplement 2C*, *Figure 3—figure supplement 3D*, *Figure 3—figure supplement 4D–E*, *Figure 4D–E*, and *Figure 5—figure supplement 1F-G*.

• Supplementary file 2. Motif analysis.This file contains enrichment values (logP) and frequencies of known motifs from HOMER in the following groups of loci: OD, CD, SD, OCT4 OFF upregulated (OCT4up), SOX2 OFF upregulated (SOX2up), and clusters 1–4. Only motifs with -logP < 50 in at least one group are shown.

• Supplementary file 3. Random forest model results.This file contains the top 500 features of the random forest model used to predict the cluster of regions based on overlap with ChIP-seq peaks from cistromeDB annotated as belonging to mouse ES cells. Importance values are derived from the model. Sample name, Factor, Cell line, and GSMID refer to sample data in GEO and ID refers to the sample ID in cistromeDB. Cluster 1–4 columns indicate fraction of regions overlapping the sample peaks.

• Supplementary file 4. Primers used for RT-qPCR.This file contains the oligonucleotide sequences used to perform RT-qPCR experiments.

• Supplementary file 5. Published datasets used.This file contains descriptions of publicly available raw data that were aligned and processed according to the Materials and methods section as well as publicly available pre-processed data used in the study.

• Transparent reporting form

### Data availability
Sequencing data have been deposited in GEO under accession code GSE134680. Source data have been provided for Figures 4B-C.

The following dataset was generated:

| Author(s) | Year | Dataset title | Dataset URL | Database and Identifier |
|---|---|---|---|---|
| Friman ET, Suter DM | 2019 | Dynamic regulation of chromatin accessibility by pluripotency transcription factors across the cell cycle | https://www.ncbi.nlm.nih.gov/geo/query/acc.cgi?acc=GSE134680 | NCBI Gene Expression Omnibus, GSE134680 |

The following previously published datasets were used:

| Author(s) | Year | Dataset title | Dataset URL | Database and Identifier |
|---|---|---|---|---|
| King HW, Klose RJ | 2017 | The pioneer factor OCT4 requires | https://www.ncbi.nlm. | NCBI Gene |

| | | | | | |
|---|---|---|---|---|---|
| | | | the chromatin remodeller BRG1 to support gene regulatory element function in mouse embryonic stem cells | nih.gov/geo/query/acc.cgi?acc=GSE87822 | Expression Omnibus, GSE87822 |
| Savatier P, Aksoy I | | 2014 | Genome-wide mapping of Klf4 and Klf5 in pluripotent mouse embryonic stem cells (ESCs) | https://www.ncbi.nlm.nih.gov/geo/query/acc.cgi?acc=GSE49848 | NCBI Gene Expression Omnibus, GSE49848 |
| Xiong J, Zhang Z, Chen J, Xu Y, Ding X, Nishinakamura R, Xu G, Chen S, Gao S, Zhu B | | 2016 | Cooperative Action Between SALL4A and TET Proteins in Stepwise Oxidation of 5-Methylcytosine | https://www.ncbi.nlm.nih.gov/geo/query/acc.cgi?acc=GSE57700 | NCBI Gene Expression Omnibus, GSE57700 |
| Kim K, Tanaka Y, Park I | | 2018 | Uhrf1 regulates active transcriptional marks at bivalent domains in pluripotent stem cells through Setd1a | https://www.ncbi.nlm.nih.gov/geo/query/acc.cgi?acc=GSE113915 | NCBI Gene Expression Omnibus, GSE113915 |
| Chronis C, Fiziec P, Papp B, Butz S, Bonora G, Sabri S, Ernst J, Plath K | | 2017 | Cooperative binding of Oct4, Sox2, and Klf4 with stage-specific transcription factors orchestrates reprogramming | https://www.ncbi.nlm.nih.gov/geo/query/acc.cgi?acc=GSE90895 | NCBI Gene Expression Omnibus, GSE90895 |
| Cao K, Collings CK, Shilatifard A | | 2018 | An Mll4/COMPASS-Lsd1 epigenetic axis governs enhancer function and pluripotency transition in embryonic stem cells | https://www.ncbi.nlm.nih.gov/geo/query/acc.cgi?acc=GSE99022 | NCBI Gene Expression Omnibus, GSE99022 |
| Rickels RA, Collings CK, Shilatifard A | | 2017 | Histone H3K4 monomethylation catalyzed by Trr and mammalian COMPASS-like proteins at enhancers is dispensable for development and viability | https://www.ncbi.nlm.nih.gov/geo/query/acc.cgi?acc=GSE95781 | NCBI Gene Expression Omnibus, GSE95781 |
| Kumar V, Masafumi M, Prabhakar S | | 2016 | Histone acetylation H2BK20ac marks cell-state specific active regulatory elements | https://www.ncbi.nlm.nih.gov/geo/query/acc.cgi?acc=GSE72886 | NCBI Gene Expression Omnibus, GSE72886 |
| Ishiuchi T, Sato T, Ohishi H | | 2019 | Zfp281 shapes the transcriptome of trophoblast stem cells and is essential for placental development | https://www.ncbi.nlm.nih.gov/geo/query/acc.cgi?acc=GSE111824 | NCBI Gene Expression Omnibus, GSE111824 |
| Adachi K, Nikaido I, Ura H, Ueda HR, Niwa H | | 2014 | ChIP-seq anlysis of Sox2, Tfap2c, and Cdx2 in trophoblast stem cells | https://www.ncbi.nlm.nih.gov/geo/query/acc.cgi?acc=GSE51511 | NCBI Gene Expression Omnibus, GSE51511 |
| Liu Z, Kraus WL | | 2017 | Genome-wide maps of histone marks, Sox2, Oct4 and PARP-1 in mES cells | https://www.ncbi.nlm.nih.gov/geo/query/acc.cgi?acc=GSE74112 | NCBI Gene Expression Omnibus, GSE74112 |
| Suter DM | | 2016 | Genome-wide analysis of SOX2 binding in asynchronous and mitotic mouse embryonic stem cells | https://www.ncbi.nlm.nih.gov/geo/query/acc.cgi?acc=GSE89599 | NCBI Gene Expression Omnibus, GSE89599 |
| Laue ED | | 2017 | 3D structures of individual mammalian genomes reveal principles of nuclear organization | https://www.ncbi.nlm.nih.gov/geo/query/acc.cgi?acc=GSE80280 | NCBI Gene Expression Omnibus, GSE80280 |
| Lim B, Tam W, Orlov YL | | 2009 | Genome-wide maps of Tbx3 binding sites in mouse ESCs | https://www.ncbi.nlm.nih.gov/geo/query/acc.cgi?acc=GSE19219 | NCBI Gene Expression Omnibus, GSE19219 |
| Kim J, Beck S | | 2014 | CpG island-mediated global gene regulatory modes in mouse embryonic stem cells | https://www.ncbi.nlm.nih.gov/geo/query/acc.cgi?acc=GSE48666 | NCBI Gene Expression Omnibus, GSE48666 |
| Sejr Hansen A, Cattoglio C, Pustova I, Tjian R, Darzacq X | | 2017 | CTCF and cohesin regulate chromatin loop stability with distinct dynamics | https://www.ncbi.nlm.nih.gov/geo/query/acc.cgi?acc=GSE90994 | NCBI Gene Expression Omnibus, GSE90994 |
| Suter DM, Friman ET | | 2019 | Endogenous fluctuations of OCT4 and SOX2 bias pluripotent cell fate decisions | https://www.ncbi.nlm.nih.gov/geo/query/acc.cgi?acc=GSE126554 | NCBI Gene Expression Omnibus, GSE126554 |

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
