## [Decision Letter]

**Acceptance summary:**

The authors explore to what extent the pioneering activity of SOX2 and OCT4 – a pair of collaborative pluripotency-linked transcription factors – overlaps and depends on each other for opening chromatin in ES cells and how this is regulated during the cell cycle. This is an important question for understanding the requirements and dynamics of pluripotency maintenance, cell differentiation and reprogramming. The study brings an unprecedented level of temporal resolution and functional hierarchy. It demonstrates the master role of OCT4 in rendering chromatin accessible to other transcription factors involved in pluripotent identity and reveals the striking dynamism and rapid reactivity of the chromatin of pluripotent cells compared to differentiated cells.

**Decision letter after peer review:**

Thank you for submitting your article "Dynamic regulation of chromatin accessibility by pluripotency transcription factors across the cell cycle" for consideration by *eLife*. Your article has been reviewed by three peer reviewers, and the evaluation has been overseen by a Reviewing Editor and Jessica Tyler as the Senior Editor. The following individuals involved in review of your submission have agreed to reveal their identity: Hamish King (Reviewer #2); Ken Zaret (Reviewer #3).

The reviewers have discussed the reviews with one another and the Reviewing Editor has drafted this decision to help you prepare a revised submission.

Summary:

The authors explore to what extent the pioneering activity of *SOX2* and OCT4 – a pair of collaborative pluripotency-linked transcription factors – overlaps and depends on each other for opening chromatin in ES cells and how this is regulated during the cell cycle. This is an important question for understanding the requirements and dynamics of pluripotency maintenance, cell differentiation and reprogramming. Using inducible systems of expression and silencing, as well as cell-cycle stage-specific degradation, the authors draw the following conclusions: 1) OCT4 and SOX2 regulate partially independent loci for chromatin accessibility, 2) OCT4 has a larger effect than SOX2, 3) the cooperative binding of OCT4 and SOX2 is itself regulated by chromatin accessibility, 4) chromatin accessibility is very dynamic and requires OCT4 to be re-established after M phase and also throughout the cell cycle, 5) depletion of OCT4 can reroute SOX2 binding to new sites and 6) TF binding and chromatin accessibility are tightly coupled in pluripotent cells, in contrast to the temporal separation between these two events previously reported in somatic (pituitary) cells. This last point may relate to the hyperdynamic aspect of pluripotent cell chromatin, allowing sensitivity and/or rapid response to developmental signals. Some of the conclusions confirm previous works, but the temporal resolution brought by the study is clearly novel and uniquely interesting. The reviewers appreciated the quality of the experimental design and the impact of the conclusions. There were also some recurrent concerns – mostly implying nuancing the conclusions – and requests for additional experiments to support the conclusions, as listed below.

Essential revisions:

1) Additional descriptive analyses of the four identified types of OCT4 target sites (clusters 1 to 4) in Figures 3-5 are needed, to allow speculation about the different properties that may define these distinct classes of binding sites. For example, do they display differences in the number of OCT4/SOX2 binding sites? Are they differentially bound (ChIP-seq signals) or do they exhibit motifs (motif analysis required) by other relevant TFs, such as bookmarking factors (ESRRB or CTCF would be of primary interest)? Do regulatory sites within a cluster tend to be in proximity/regulate the same category of genes (ontology)? Position regarding TADs? A greater depth of discussion/conjecture on this point is really required – why might these sites behave so differently?

2) There is little analysis relating chromatin accessibility with changes in expression at nearby genes (Figure 2—figure supplement 2 excepted). The functional impact of the observed (rather modest) chromatin changes should be tested at the level of gene expression, at least at a few target genes. This would be an interesting point to infer whether the different properties of the various clusters are functional or not and perhaps whether different clusters are more likely to be more or less important for maintaining pluripotency when exiting mitosis. The OCT4-AID-degron cell line would be the perfect system to examine this question and should avoid a lot of secondary effects that may occur even at 24 hours DOX treatment in the ZHBTc4 cell.

3) ATAC-seq is a non-specific assay that is largely interpreted as chromatin accessibility, which likely reflects TF binding. Since the authors are specifically looking at peaks that change in ATAC-seq signal before and after OCT4/SOX2 degradation/silencing, changes in ATAC-seq peaks may reflect TF binding loss, rather than direct change in accessibility. To address this concern, the authors should computationally parse the ATAC-seq peaks into different size classes, such as short fragments that represent TF binding regions and 180-250 bp fragments that represent nucleosomes. The authors could then plot the nucleosome fragments in the different peak lists and show that nucleosome occupancy is changing as a function of TF loss.

4) All reviewers agreed that statistical analyses are required throughout the manuscript. For example, in Figure 2—figure supplement 2B, a small decrease in ATAC-seq signal is observed at OD sites after SOX2 depletion, yet the major conclusion from the authors is that chromatin accessibility is unchanged. Are the authors arguing that this change is insignificant? Hence the need of statistical analyses.

5) Data should be presented as heatmaps more consistently, not only as bar plots (as it currently is) to really reflect the order of magnitude of the effects observed on chromatin accessibility and histone marks (which are in reality rather modest, but expanded by the bar graph displays). It is at least the case for Figure 3—figure supplement 2C-D (as was done in Figure 3—figure supplement 2B) and in Figure 4D.

---

## [Author Response]

Essential revisions:1) Additional descriptive analyses of the four identified types of OCT4 target sites (clusters 1 to 4) in Figures 3-5 are needed, to allow speculation about the different properties that may define these distinct classes of binding sites. For example, do they display differences in the number of OCT4/SOX2 binding sites? Are they differentially bound (ChIP-seq signals) or do they exhibit motifs (motif analysis required) by other relevant TFs, such as bookmarking factors (ESRRB or CTCF would be of primary interest)? Do regulatory sites within a cluster tend to be in proximity/regulate the same category of genes (ontology)? Position regarding TADs? A greater depth of discussion/conjecture on this point is really required – why might these sites behave so differently?

We agree that additional analysis can be informative about the nature of these clusters. As described in the original manuscript, we found that unaffected regions were enriched near promoters and we had identified an enrichment for OCT4-SOX2 motifs (Figure 3E) and OCT4 binding (Figure 3F) in cluster 1. We had also performed motif analysis that did not reveal other clear differences between the clusters (revised Supplementary file 2) and commented on this in the original manuscript.

We have now expanded this analysis substantially. As suggested, we analyzed the number of OCT4/SOX2 binding sites (Figure 3E), showing that cluster 1 loci have a higher prevalence of two binding sites in the same region. We also analyzed motif enrichment in the entire region (defined by peak calling), as opposed to 200bp close to the peak center as done previously, and replaced all motif analysis in the manuscript with whole region motif enrichment. This yielded mostly similar results to what we already commented on in the original manuscript. However, this analysis also yielded additional differential motif enrichments, including lower enrichment for CTCF motifs in Cluster 1 and slight enrichment for ESRRB motifs in Cluster 3. In addition, we have now used the large body of publicly available ChIP-seq datasets that have been processed by a standardized pipeline as part of the cistromeDB project to perform a more unbiased screening of potential co-binding factors in the different clusters that may not be detected by motif analysis. We analyzed the peak overlap of 3’628 ChIP-seq datasets from mouse ES cells in cistromeDB that overlapped with at least one region in our clusters and used this information to train a random forest model to classify the different clusters. The trained model predicted the correct cluster in the test data in 46.5% of cases, as opposed to 25% expected by random chance. Therefore, the features used by the model and enriched in the different clusters can inform us on factors that distinguish the clusters. Top parameters in the model included data from dox-treated ZHBTc4 cells, including a depletion of SOX2 and SS18 binding in cluster 1 after OCT4 depletion compared to the other clusters. This confirms that our approach can identify features that distinguish the clusters. The model uncovered several other expected top parameters, including promoter marks (H3K4me3, POLR2B, H3K9ac) enriched in Cluster 4 and POU5F1 (OCT4) enriched in Cluster 1. In addition, we detected several other pluripotency factors enriched in clusters 1 and 3 including NANOG, SOX2, NR0B1, and SALL4. We thus decided to check for overlap with the network of pluripotency regulators from Dunn et al., 2014, which were enriched in clusters 1 and 3. In accordance with our motif analysis, we found lower enrichment of CTCF, as well as RAD21, binding in cluster 1 compared to the other clusters. This new analysis is described in the manuscript (subsection “OCT4 is required at the M-G1 transition to re-establish enhancer accessibility”, last paragraph) and can be found as Figure 3—figure supplement 4C-E and Supplementary file 3. We have also included a statement on the relevance of these findings in the Discussion (second paragraph) in the context of pluripotency loss when OCT4 is depleted.

As suggested by the reviewers, we also looked for gene ontology of nearby genes, which revealed differential enrichment of KEGG GO categories in the different clusters, with pluripotency most enriched in cluster 1 (Figure 3—figure supplement 3B). This is described in the third paragraph of the subsection “OCT4 is required at the M-G1 transition to re-establish enhancer accessibility”. Note that we have utilized a Z-normalized scale for relative GO enrichment among the different clusters to avoid bias coming from the different number of regions, as previously done in Schwanhäusser et al. Nature 2011. We have modified all our previous GO analysis to utilize this method, which did not change any of the conclusions (see e.g. revised Figure 2G-I) and described this in the Materials and methods section. We also show that genes near clusters 1-3 are enriched for downregulation upon OCT4 depletion (Figure 3—figure supplement 3A). As suggested, we looked for the position of loci with regards to TAD boundaries (from http://chromosome.sdsc.edu/mouse/hi-c/download.html) and found a very minor enrichment in larger distance to TAD boundaries in cluster 1. Because of the small difference, we have not included this analysis in the manuscript (Author response image 1).

2) There is little analysis relating chromatin accessibility with changes in expression at nearby genes (Figure 2—figure supplement 2 excepted). The functional impact of the observed (rather modes)– chromatin changes should be tested at the level of gene expression, at least at a few target genes. This would be an interesting point to infer whether the different properties of the various clusters are functional or not and perhaps whether different clusters are more likely to be more or less important for maintaining pluripotency when exiting mitosis. The OCT4-AID-degron cell line would be the perfect system to examine this question and should avoid a lot of secondary effects that may occur even at 24 hours DOX treatment in the ZHBTc4 cell.

This is a good point. Importantly, it is not straightforward to directly compare the effect of changes in chromatin accessibility of the different clusters on transcription, due to the difficulty in assigning specific regulatory elements to their corresponding gene. Based on chromosome conformation capture assays, only 47% of distal regulatory elements interact with the nearest expressed TSS (Sanyal et al., Nature 2012). Furthermore, we do not expect large changes in transcription because of the relatively small changes in chromatin accessibility inherent to our experimental system.

Nevertheless, we annotated the closest OCT4-bound region to each gene (excluding those where the region was <1 kb from TSS) and among these and for Clusters 1-3, selected 5 genes for which expression was significantly downregulated upon OCT4 depletion for 24 hours according to King et al., 2017. As controls, we chose 7 genes closest to Cluster 4 regions that were shown unaffected by OCT4 depletion (also King et al., 2017). We extracted RNA from 5 replicates of mCherry-OCT4-AID cells untreated or treated with IAA for 2 hours and performed RT-qPCR using primers flanking intron-exon junctions and exon-exon junctions. Pre-mRNAs of genes close to elements in cluster 1-2 and some in cluster 3 tended to be modestly downregulated upon OCT4 depletion, while those close to cluster 4 regions or mature mRNAs were generally not (Figure 5—figure supplement 1F-G). This suggests that transcription of genes near loci whose accessibility depends on OCT4 can be affected by acute OCT4 depletion. This new analysis is included in the last paragraph of the subsection “Dynamic relationship between OCT4 concentration and chromatin accessibility”.

3) ATAC-seq is a non-specific assay that is largely interpreted as chromatin accessibility, which likely reflects TF binding. Since the authors are specifically looking at peaks that change in ATAC-seq signal before and after OCT4/SOX2 degradation/silencing, changes in ATAC-seq peaks may reflect TF binding loss, rather than direct change in accessibility. To address this concern, the authors should computationally parse the ATAC-seq peaks into different size classes, such as short fragments that represent TF binding regions and 180-250 bp fragments that represent nucleosomes. The authors could then plot the nucleosome fragments in the different peak lists and show that nucleosome occupancy is changing as a function of TF loss.

We thank the reviewers for this suggestion. We have now split the reads into bins of 0-100bp (nucleosome-free regions) and 180-250bp (single nucleosomes flanked by accessible regions). We decided to perform this analysis on the acute degradation of OCT4 using IAA treatment for 2 hours as this allows us to clearly assess the direct effects of TF loss on chromatin accessibility. Figure 5—figure supplement 1A-E show that the loss in accessibility upon acute OCT4 depletion also affects nucleosomal regions and therefore excludes that changes in accessibility simply reflect loss of TF binding. We have modified the text to incorporate this analysis (see the last paragraph of the subsection “Dynamic relationship between OCT4 concentration and chromatin accessibility”).

4) All reviewers agreed that statistical analyses are required throughout the manuscript. For example, in Figure 2—figure supplement 2B, a small decrease in ATAC-seq signal is observed at OD sites after SOX2 depletion, yet the major conclusion from the authors is that chromatin accessibility is unchanged. Are the authors arguing that this change is insignificant? Hence the need of statistical analyses.

We agree that more thorough statistical analysis of our data is useful. We have now included null hypothesis testing using either t-tests or Mann-Whitney U/Wilcoxon tests (when Shapiro-Wilk’s showed evidence of non-normality) at all comparisons of distributions (see e.g. Figure 3—figure supplement 1D, E, G, H, I, J). We have also included Fisher’s exact tests for comparisons of categories (see e.g. Figure 2—figure supplement 2E). Furthermore, we have displayed the p-values of all motif enrichments in the plots. We have justified and explained our choice of statistical tests in the Materials and methods section.

Note that concerning the example mentioned above (Figure 2—figure supplement 2B), these loci have been statistically tested for significant difference in accessibility using the limma algorithm at the differential analysis stage (Figure 1C). However, we did not want to claim that there is no statistically significant difference when analyzing these data as a group, but rather that the effect size is small. We have now included statistical analysis of the distribution of values used to derive metaplots (such as the one referred to) where comparisons are made (see Supplementary file 1). This reveals that there is indeed a strong statistical significance at all comparisons due to the large number of data points measured. We have been careful in our writing so that our claims do not conflate effect size and significance.

5) Data should be presented as heatmaps more consistently, not only as bar plots (as it currently is) to really reflect the order of magnitude of the effects observed on chromatin accessibility and histone marks (which are in reality rather modest, but expanded by the bar graph displays). It is at least the case for Figure 3—figure supplement 2C-D (as was done in Figure 3—figure supplement 2B) and in Figure 4D.

We agree that heatmaps are informative to appreciate that while differences can be observed globally, they are rather small using our experimental system (as we also commented on in the Discussion). We have included the suggested heatmaps in revised Figure 3—figure supplement 2E and Figure 4—figure supplement 2B, D and Figure 5—figure supplement 1E. Because it is difficult to visually appreciate these differences using such plots, we have also included heatmaps of fold-change values (Figure 3—figure supplement 2F, Figure 4—figure supplement 2E).